# COMPOSING ENSEMBLES OF PRE-TRAINED MODELS VIA ITERATIVE CONSENSUS

**Shuang Li** \*†
†MIT CSAIL
lishuang@mit.edu

**Yilun Du**†
MIT CSAIL
yilundu@mit.edu

**Joshua B. Tenenbaum**
MIT CSAIL, BCS, CBMM
jbt@mit.edu

**Antonio Torralba**
MIT CSAIL
torralba@mit.edu

**Igor Mordatch**
Google Brain
imordatch@google.com

## ABSTRACT

Large pre-trained models exhibit distinct and complementary capabilities dependent on the data they are trained on. Language models such as GPT-3 are capable of textual reasoning but cannot understand visual information, while vision models such as DALL-E can generate photorealistic photos but fail to understand complex language descriptions. In this work, we propose a unified framework for composing ensembles of different pre-trained models – combining the strengths of each individual model to solve various multimodal problems in a zero-shot manner. We use pre-trained models as "generators" or "scorers" and compose them via closed-loop iterative consensus optimization. The generator constructs proposals and the scorers iteratively provide feedback to refine the generated result. Such closed-loop communication enables models to correct errors caused by other models, significantly boosting performance on downstream tasks, *e.g.* improving accuracy on grade school math problems by $7.5\%$, without requiring any model finetuning. We demonstrate that consensus achieved by an ensemble of scorers outperforms the feedback of a single scorer, by leveraging the strengths of each expert model. Results show that the proposed method can be used as a general purpose framework for a wide range of zero-shot multimodal tasks, such as image generation, video question answering, mathematical reasoning, and robotic manipulation.

## 1 INTRODUCTION

Large pre-trained models have shown remarkable zero-shot generalization abilities, ranging from zero-shot image generation and natural language processing to machine reasoning and action planning. Such models are trained on large datasets scoured from the internet, often consisting of billions of datapoints. Individual pre-trained models capture different aspects of knowledge on the internet, with language models (LMs) capturing textual information in news, articles, and Wikipedia pages, and visual-language models (VLMs) modeling the alignments between visual and textual information. While it is desirable to have a single sizable pre-trained model capturing all possible modalities of data on the internet, such a comprehensive model is challenging to obtain and maintain, requiring intensive memory, an enormous amount of energy, months of training time, and millions of dollars. A more scalable alternative approach is to compose different pre-trained models together, leveraging the knowledge from different expert models to solve complex multimodal tasks.

Building a unified framework for composing multiple models is challenging. Prior works (Alayrac et al., 2022; Zeng et al., 2022) have explored composing pre-trained models in two main ways: (jointly) finetuning models on large datasets, or using common interfaces such as language to combine

---

\*Correspondence to: Shuang Li <lishuang@mit.edu>.

†indicates equal contribution. Shuang Li did all the experiments on image generation, video question answering, and mathematical reasoning. Yilun Du did all the experiments on robot manipulation.

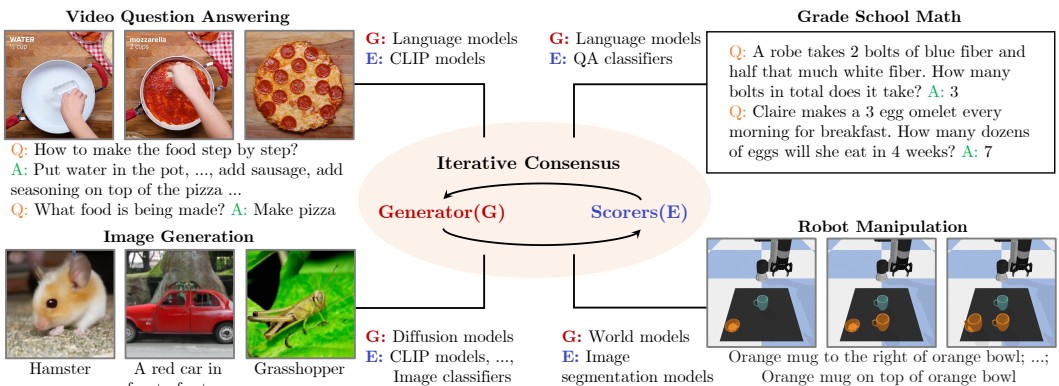

Figure 1: The proposed framework that composes a "generator" and an ensemble of "scorers" through iterative consensus enables zero-shot generalization across a variety of multimodal tasks.

different models. However, these works have several key limitations: First, simply combining models does not fully utilize each pre-trained model as there is no closed-loop feedback between models. Cascading models, such as Socratic models (Zeng et al., 2022), allows one-way communication but prevents information processed by later models from propagating back to earlier models to correct errors. Secondly, common interfaces are limited to particular types of models. Language is used as the intermediate connection in Socratic models (Zeng et al., 2022), but a language interface is insufficient to solve many real-world tasks, such as continuous robot control, which requires continuous representations. In addition, Socratic models require pre-designed language templates for the communication between models, which limits scalability. Thirdly, jointly finetuning multiple models (Alayrac et al., 2022) requires careful optimization to ensure that the model behaviors remain stable. Such models also require intensive memory and large datasets and can only be used for solving specific tasks.

To resolve these difficulties, we propose a unified framework to compose models in a zero-shot manner[1] without any training/finetuning. Our framework employs a single model as a generator and an ensemble of scorers. The generator iteratively generates proposals, and each scorer provides a feedback score indicating their agreement. The generator refines its outputs until all the scorers achieve a final consensus. This iterative closed-loop communication between the generator and scorers enables models to correct the errors caused by other models, substantially boosting performance.

The ensemble of scorers is inspired by the idea of "wisdom of the crowds". Each scorer provides complementary feedback to the generator, compensating for the potential weaknesses of other scorers. A Vision-Language scorer, for example, may correct the biases of a language model. We notice that different pre-trained model instances from the same family have diversity of outputs, which leads to more robust scorers. We demonstrate that guiding the generator with such an ensemble of scorers significantly outperforms a generator guided by a single scorer.

To summarize, our work has three main contributions.

- First, we propose a unified framework for composing pre-trained models across a variety of tasks, such as image generation, video question answering, mathematical reasoning, and robot manipulation.

- Second, we illustrate how the proposed framework can effectively solve zero-shot multimodal tasks without any training/finetuning. The closed-loop communication between the generator and scorers allows the models to interact with each other to improve performance iteratively.

- Finally, we illustrate how our framework enables the use of ensembles of different pre-trained models as scorers, significantly improving the zero-shot results by leveraging the strengths of multiple expert models.

These observations point to the effectiveness of the proposed method as a general purpose framework for composing pre-trained models for solving various zero-shot multimodal tasks.

---

[1]By zero-shot, we mean the composed models are never trained together on the evaluation task.

## 2 RELATED WORK

Large pre-trained models have shown great success across a variety of domains, such as language generation/translation, image generation, and decision-making.

**Language models.** Large language models, such as ELMo (Peters et al., 2018), BERT (Devlin et al., 2018), and GPT-2 (Radford et al., 2019), are able to achieve state-of-the-art performance on many standard NLP benchmarks. More recent works, such as GPT-3 (Brown et al., 2020), PALM (Chowdhery et al., 2022), and Chinchilla (Hoffmann et al., 2022) further enable few-shot learning from textual prompts. **Vision-language models.** Large pre-trained vision-language generative models, such as DALL-E 2 (Ramesh et al., 2022), Parti (Yu et al., 2022), and Imagen (Saharia et al., 2022), can generate high-resolution images given natural language descriptions. Large pre-trained vision-language discriminative models, such as CLIP (Radford et al., 2021), convert images and languages into the same feature space, achieving remarkable zero-shot generalization ability on downstream tasks. **Decision-making models.** Large pre-trained models have been widely applied to solve decision-making tasks, such as learning general purpose policies (Reed et al., 2022; Li et al., 2022; Shridhar et al., 2022), making planners (Huang et al., 2022; Ahn et al., 2022), and learning world models (Ebert et al., 2018). However, due to the large variability in decision-making tasks, no existing pre-trained models can be readily applied across different tasks.

**Composing pre-trained models.** Composing large pre-trained models has been widely studied recently. The predominant way to compose pre-trained models is to (joint) finetune them on new tasks (Li et al., 2019; Wang et al., 2021; Alayrac et al., 2022; Mokady et al., 2021), but such approaches are computationally expensive. Alternative approaches compose models through a common interface such as language(Tewel et al., 2021; Zeng et al., 2022). Other works compose pre-trained models by composing learned probability distributions of the data, such as energy-based models (Liu et al., 2022; 2021; Du et al., 2020), which can be applied to generate images or to refine structural prediction (Belanger et al., 2017). In this paper, we propose a general framework to compose pre-trained models across a variety of domains without any training or finetuning.

## 3 METHOD

Given a set of large pre-trained models, we aim to utilize the expert knowledge from different models to solve zero-shot multimodal tasks. We separate pre-trained models into two categories – generators ($G$) such as GPT (Brown et al., 2020; Radford et al., 2019) and Diffusion models (Ho et al., 2020) that can generate candidate solutions, and scorers ($E$) such as CLIP (Radford et al., 2021) and classifiers that output a scalar score to evaluate each generated solution. We propose **PIC** (composing ensembles of **P**re-trained models via **I**terative **C**onsensus), a framework which composes ensembles of pre-trained models for multimodal tasks. The core idea of PIC is to generate solutions through iterative optimization, where we leverage the knowledge from different models to jointly construct a consensus solution. In PIC, a generator $G$ iteratively and sequentially generate candidate solutions, each of which is refined based on the feedback from a set of scorers. In particular, we seek to obtain a solution $x^*$ such that

$$x^* = \arg\min_{x \sim G} \sum_n E_n(x), \tag{1}$$

where $\{E_n\}$ is the set of scorers. At each iteration, we refine the solutions to have a lower score than the previous iterations. This procedure, described in Equation (1), converges to a solution that minimizes the energy across multiple pre-trained models, which maximizes the agreement between the generator and scorers. In contrast to Socratic Models where different pre-trained models are called sequentially, the closed-loop iterative refinement through which we obtain $x^*$ enables the generator and scorers to communicate with each other to reach a consensus on the final solution.

Below, we illustrate how PIC can be broadly applied across tasks in image generation, video question answering, grade school math, and robot manipulation. To optimize Equation (1), we consider two different optimization procedures – either a continuous approach that leverages the gradients of each scorer $E_n(x)$ or a discrete approach that directly samples possible solutions.

### 3.1 APPLICATIONS TO ZERO-SHOT TASKS

**Image generation.** We first apply the proposed framework to image generation to generate images conditioned on a text description or a class label. We use the reverse diffusion process of

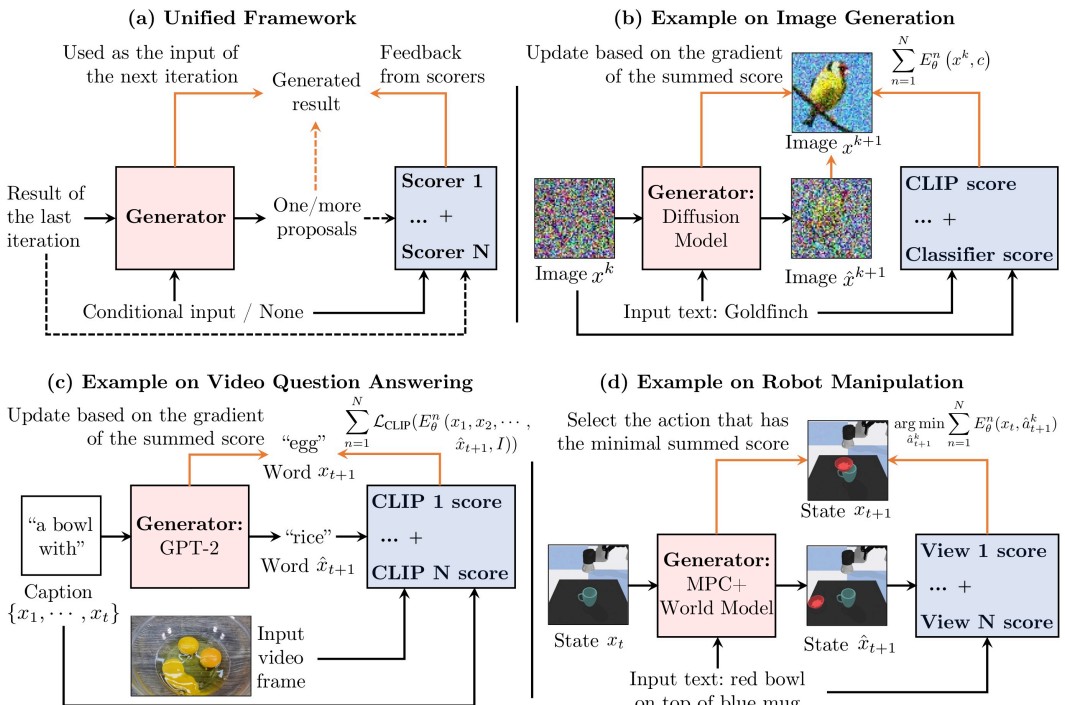

Figure 2: **The proposed unified framework and examples on three representative tasks.** (a) Overview of the proposed unified framework. Dashed lines are omitted for certain tasks. (b) Image generation. A pre-trained diffusion model is used as the generator, and multiple scorers, such as CLIP and image classifiers, are used to provide feedback to the generator. (c) Video question answering. GPT-2 is used as the generator, and a set of CLIP models are used as scorers. (d) Robot manipulation. MPC+World model is used as the generator, and a pre-trained image segmentation model is used to compute the scores from multiple camera views to select the best action. Orange lines represent the components used to refine the generated result.

GLIDE (Nichol et al., 2021), a text-guided diffusion model, as the generator to generate image proposals. At each step of the diffusion process (corresponding to a step of the iterative refinement), we use the gradient from an ensemble of scorers, such as CLIP (Radford et al., 2021), to guide and update the generated proposals. We iteratively repeat this procedure until the final step.

As shown in Fig. 2 (b), the image $x^k$ generated at iteration $k$ is first sent to the diffusion model to generate an image proposal $\hat{x}^{k+1}$. Each scorer outputs a score to evaluate whether the generated image matches the given text input. For example, CLIP computes the cosine similarity between the image and text features as the score. The scores generated by different scorers are summed, and their gradient with respect to $x^k$ is used to compute the next reverse prediction $x^{k+1}$:

$$x^{k+1} \leftarrow \hat{x}^{k+1} + \lambda \nabla_{x^k} \sum_{n=1}^{N} E_\theta^n \left( x^k, c \right), \tag{2}$$

where $N$ is the number of scorers and $c$ is the text label. We denote the reverse process prediction as $x^{k+1}$ instead of $x^{k-1}$ (used by most diffusion models) to keep the consistent notation across tasks.

**Video question answering (VQA).** Caption generation for a single video frame is shown in Fig. 2 (c). We use GPT-2 as the generator and multiple different CLIP models, trained with different configurations, as the scorers. Given a video frame $I$, we generate a sequence of words to describe it. To integrate feedback from scorers to the generator, similar to (Tewel et al., 2021), we define a context cache $C_t$ (a set of embedding functions in GPT-2) that stores the context information generated so far, which is updated iteratively based on the feedback from scorers. The prediction of the next word from the generator $G$ is given by $x_{t+1} = G(x_t, C_t)$. To update $C_t$, we first use $G$ to generate a set of candidate words $\hat{X}_{t+1} = \{\hat{x}_{t+1}\}$, and then use the feature distance (after softmax) between each sentence (the concatenation of previous words and each new word $\{x_1, x_2, \cdots, \hat{x}_{t+1}\}$, where $\hat{x}_{t+1} \in \hat{X}_{t+1}$) and the video frame as the probability of them matching. The CLIP score is the cross-entropy loss $\mathcal{L}_{\text{CLIP}}$ between this new probability distribution and the original distribution of the

next word obtained from the generator $G$. The gradient of the summed score (multiple CLIP models) is then propagated to $G$ to update $C_t$:

$$C_t^{k+1} \leftarrow C_t^k + \lambda \nabla_{C_t^k} \sum_{n=1}^{N} \mathcal{L}_{\text{CLIP}}(E_\theta^n (x_1, x_2, \cdots, \hat{x}_{t+1}, I)), \tag{3}$$

where $k$ is the step of iterative refinement. After several iterations, the updated $C_t$ is used to generate the next token $x_{t+1} = G(x_t, C_t)$. We repeat this process until we generate the entire caption. We cascade the captions of multiple video frames and questions about this video to prompt GPT-3 for video question answering (See Appendix B.2).

**Grade school math.** We further apply PIC to solve grade school math problems. We use GPT-2 as the generator and treat the grade school math problem as a text generation problem. The scorer, a pre-trained question-solution classifier, provides the generator feedback to guide the next token's generation $x_{t+1}$. We follow the approach used in VQA to iteratively optimize the generations based on the feedback from scorers. Our generator $G$ first generates a set of candidate words $\hat{X}_{t+1} = \{\hat{x}_{t+1}\}$, and then the classifier predicts the probability of each solution (the concatenation of previous words and each new word $\{x_1, x_2, \cdots, \hat{x}_{t+1}\}$, where $\hat{x}_{t+1} \in \hat{X}_{t+1}$) matching the given question. The classifier score is the cross-entropy loss between this new probability distribution and the original distribution of the next word obtained from the generator $G$. The gradient of the classifier score is used to update $C_t$ through iterative refinement, same as Eq. (3). The updated $C_t$ is used to predict the next word $x_{t+1} = G(x_t, C_t)$. We repeat this process until we generate the complete solution.

**Robot manipulation.** Finally, we illustrate how PIC can be applied to manipulate objects in the robot environment to conform to a set of object relations such as "red bowl on top of blue mug" shown in Fig. 2 (d). We use the combination of Model Predictive Control (MPC) (Williams et al., 2015) and the World Model as the generator. At each time step, we first use MPC to sample a set of possible actions and then render the state images (after executing an action) from multiple camera views using the world model. For each action, the scorer computes a summed score across all camera views as its final score, which is used to select the best action to execute. Thus, in this domain, the ensemble consists of scorers based on different views of the scene.

For the generator, we assume that there is a pre-trained model, *i.e.* world model, that can accurately render and simulate the dynamic changes in the robot world. Since such a large pre-trained model does not directly exist, we approximate it using an environment simulator combined with MPC as the generator. For the scorer, we use the pre-trained ViLD (Gu et al., 2021) to generate segmentation maps for images captured by different camera views $n$, and the corresponding text label for each segment, which are used to obtain object relations. We compare the generated object relations and the relations specified by the text description to obtain the score, *i.e.* score equals 0 if they match; otherwise, 1 (here the score means the distance) (see Appendix B.4 for details). To obtain a final world state $x_T$ that satisfies the specified relations, and the action sequence $\{a_1, \cdots, a_T\}$ that manipulates the objects into the final state $x_T$, the generator iteratively samples possible actions $\hat{a}_{t+1}^k$ and gets feedback from scorers. The best action is selected as:

$$a_{t+1} = \arg\min_{\hat{a}_{t+1}^k} \sum_{n=1}^{N} E_\theta^n (x_t, \hat{a}_{t+1}^k). \tag{4}$$

Each scorer, $E_\theta^n$, outputs a score for the resultant state obtained when a candidate action $\hat{a}_{t+1}^k$ is applied to the current world state $x_t$. We execute $a_{t+1}$ in the environment and get a new state $x_{t+1}$. We repeat this process until the task is accomplished or we are at the final step $T$.

## 4    EXPERIMENT SETUP

We evaluate the proposed framework for composing pre-trained models on four representative tasks, including image generation, video question answering, grade school math, and robot manipulation.

**Image generation.** We first show that composing the pre-trained image generator and scorer models such as CLIP enables effective zero-shot image generation. We evaluate the image generation results on ImageNet (Deng et al., 2009) with the image resolution of $64 \times 64$. The class labels are used as the text input to guide image generation. Each method generates 50 images for each class. We evaluate the image generation quality using Inception Score (IS) (Salimans et al., 2016), Fréchet

Table 1: **Image generation results on ImageNet.** Our PIC can compose the pre-trained generator (G) and scorers (E) through iterative optimization. Composing multiple scorers further boosts performance.

| Method Name | Generator | Scorer | IS ↑ | FID ↓ | KID ↓ |
|---|---|---|---|---|---|
| PIC (G+E1) | GLIDE | CLIP | 25.017 | 30.462 | 6.174 |
| PIC (G+E2) | GLIDE | CLS | 22.077 | 30.871 | 7.952 |
| PIC (G+E3) | GLIDE | CLS-FREE | 25.926 | 29.219 | 5.325 |
| PIC (G+E1+E2+E3) | GLIDE | CLIP + CLS + CLS-FREE | **34.952** | **29.184** | **3.766** |

Table 2: **Video question answering results on ActivityNet-QA.** JustAsk (FT) is finetuned on ActivityNet-QA, thus achieving the best results. For zero-shot VQA, our method (PIC) significantly outperforms JustAsk (Pretrain), one of the best VQA methods. Using multiple scorers further improves the performance.

| Method Name | Zero-Shot | Generator | Scorer | Accuracy ↑ | Vocab ↑ |
|---|---|---|---|---|---|
| JustAsk (FT) | No | - | - | **64.667** | 160 |
| JustAsk (Pretrain) | Yes | - | - | 50.671 | 210 |
| PIC (G+E1) | Yes | GPT-2 | CLIP-32 | 58.389 | 267 |
| PIC (G+E1+E2+E3) | Yes | GPT-2 | CLIP-32 + CLIP-14 + CLIP-multilingual | **61.168** | **304** |

Inception Distance (FID) (Heusel et al., 2017), and Kernel Inception Distance (KID) (Bińkowski et al., 2018). IS measures the distribution of generated images. Higher values mean the models can generate more distinct images. FID considers the distributions of both generated images and real images. Lower scores represent that the generated images are closer to the real images. KID is similar to FID, measuring the similarity between two data distributions, but is in the kernel space.

**Video question answering.** We evaluate methods for solving VQA tasks on ActivityNet-QA (Yu et al., 2019). Our method generates free-form language answers instead of selecting an answer from a pre-defined answer set (Yang et al., 2021; Lei et al., 2022). To evaluate such free-form VQA, we ask workers from Amazon Mechanical Turk to measure whether the generated answer matches the given question and video (See Appendix C for IRB approval and experimental details). For fair comparisons, all the approaches answer the same 300 video questions, and each answer is evaluated by three different workers. The accuracy rate and vocabulary size are reported. An answer is correct if at least two workers believe it is correct. The accuracy rate is the percentage of correctly answered questions over all the questions. To evaluate the diversity of generated answers, we also report the vocabulary size (*i.e.* the number of words) of answers generated by each method.

**Grade school math.** GSM8K (Cobbe et al., 2021) is a dataset for grade school math problems. Each problem consists of a question, intermediate analyses, and a final solution. We evaluate approaches to solving problems on the 1K test set. We use beam search to generate candidate solutions. The accuracy of beam size 1 and beam size 5 are reported. For beam size of 1, we mark the result as correct if it matches the final solution. For beam size of 5, we mark the result as correct if any of the five generated results matches the solution.

**Robot manipulation.** We next evaluate how pre-trained models may be used to manipulate objects in Ravens (Zeng et al., 2020). In Ravens, the action space of robot is to drop an object at a 2D location on the table. The goal is to obtain a scene configuration that satisfies the object relations specified by a textual description or a real-world image, such as "blue mug to the left of purple bowl". The task is successful if the object relations in the final state satisfy all the relations specified by the input text or image. We report the success rate of tasks with two and three specified object relations.

## 5 EXPERIMENTS

We compare the proposed method with baselines on the above four zero-shot tasks.

### 5.1 IMAGE GENERATION

We evaluate the zero-shot conditional image generation on ImageNet in Table 1. We first show results of composing a single generator (G) and a single scorer (E). We compose GLIDE (Nichol et al., 2021) with three different types of scorers, respectively. E1 is CLIP (Radford et al., 2021) that computes the cosine similarity between the image and text features as the score, E2 is the image classifier (CLS) (Dhariwal & Nichol, 2021) that predicts the probability of the image matching the text label as the score, and E3 is the classifier-free guidance (CLS-FREE) (Ho & Salimans, 2022)

| One video frame | | JustAsk (Pretrain) | PIC (G+E1) | PIC (G+E1+E2+E3) | JustAsk (FT) (not zero-shot) |
|---|---|---|---|---|---|
| | Q: is the person in blue a man or a woman? | A: no | A: woman | A: woman | A: yes |
| | Q: what happened before the pole vault? | A: audience cheered | A: the person in the video is stretching | A: the athlete is running | A: magnesia powder |
| | Q: how many people are there in the video? | A: no | A: 1 | A: 4 | A: 2 |
| | Q: what kind of trousers does the woman wearing yellow clothes look like? | A: tight trousers | A: short | A: short | A: short |
| | Q: what is the person with hat doing? | A: nursing bicycle | A: using a pickup | A: cutting grass | A: weed |
| | Q: what happened to the person in the hat before the engine? | A: someone passed through | A: put on the helmet | A: turn on the engine | A: speech |

Figure 3: **Video question answering example results.** Our approach successfully identifies gender and clothing, but its failure to count objects is a reflection of GPT-2 and CLIP's inability to count.

which can be treated as an implicit classifier that directly provides pixel-wise gradient feedback to the generated image (Appendix B.1). We then compose the generator with all scorers, *i.e.* G+E1+E2+E3. Composing the generator and a single scorer allows zero-shot image generation. Composing multiple scorers significantly outperforms a single scorer. We note that the generator is not trained on ImageNet; thus the results in Table 1 cannot be directly compared with methods trained on ImageNet.

## 5.2 VIDEO QUESTION ANSWERING

**Quantitative results.** We compare PIC with one of the state-of-the-art VQA approaches, *i.e.* JustAsk (Yang et al., 2021), on ActivityNet-QA (Yu et al., 2019). In Table 2, JustAsk (FT) is finetuned on ActivityNet-QA, thus achieving the best results. We then compare PIC with JustAsk (Pretrain) for zero-shot VQA. The generator of our method, GPT-2 (medium size), is trained on Webtext (Radford et al., 2019) using the Huggingface library (Wolf et al., 2019). Our scorers are CLIP models (Radford et al., 2021; Reimers & Gurevych, 2019) trained on different datasets or using different configurations. PIC (G+E1) outperforms JustAsk (Pretrain) by %7.72. Composing more scorers further improves the accuracy by %2.78. In addition, the vocabulary size of answers generated by our method is larger than other approaches, indicating that our method can answer questions using richer language and more diverse phrasing. Note that our method solves a "more challenging" problem than JustAsk (Pretrain) and JustAsk (FT). Our method generates open-language answers while JustAsk (Pretrain) and JustAsk (FT) select an answer from a pre-defined answer set. Generating free-form responses requires both semantic and grammatical correctness. PIC performs well on both these dimensions while also using a richer vocabulary.

**Qualitative results.** In Fig. 3, we show answers generated by different approaches given a video (only showing a single video frame) and questions. Our approach successfully identifies gender and clothing, but none of the approaches know how to count numbers.

## 5.3 GRADE SCHOOL MATH

**Quantitative results.** In Table 3, we compare PIC with two baselines, *i.e.* GPT-Pretrain and GPT-FT, for solving math problems on GSM8K (Cobbe et al., 2021). GPT-Pretrain uses the pre-trained GPT-2 (medium size GPT-2 trained on Webtext using Huggingface) to generate numeric strings. GPT-FT is based on GPT-Pretrain and then finetuned on GSM8K. Our method

Table 3: **Grade school math results on GSM8K.** Our method (PIC) that composes GPT-2 and a pre-trained question-solution classifier significantly outperforms the baselines, including GPT-FT that is finetuned on GSM8K.

| Method Name | Generator | Scorer | BS=1 ↑ | BS=5 ↑ |
|---|---|---|---|---|
| **GPT-Pretrain** | GPT-2 (Pretrain) | - | 1.744 | 12.206 |
| **GPT-FT** | GPT-2 (FT) | - | 3.487 | 18.271 |
| **PIC (G+E)** | GPT-2 (Pretrain) | CLS | **16.831** | **20.773** |

uses the same GPT-2 (Pretrain) as the generator and a question-solution classifier (CLS) as the scorer. The classifier is trained on GSM8K to distinguish whether a solution is correct for a given question. We surprisingly find that PIC achieves significantly better performance than GPT-FT (%13.344 higher on beam size 1), even though the generator has never seen the math problems before. The classifier

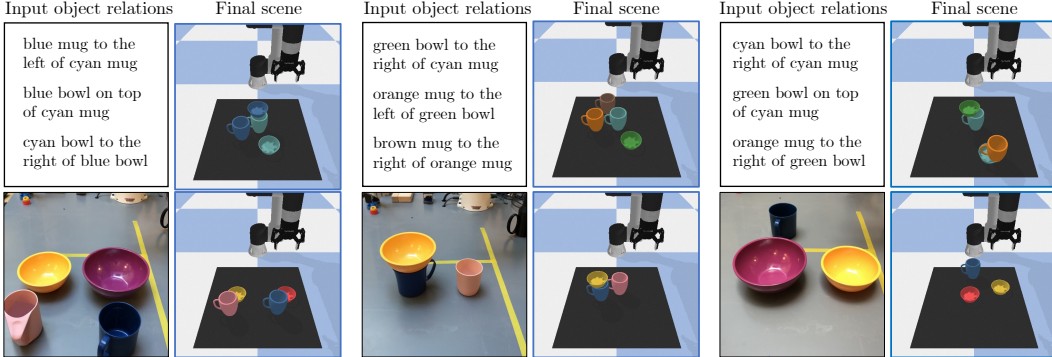

Figure 4: **Robot manipulation example results.** The robot manipulates objects to achieve certain object relations that are specified by textual descriptions (first row) or real-world images (second row).

only provides feedback to the generator, but through iterative refinement, combining a generator and a scorer without joint training is more effective than directly finetuning GPT-2 on GSM8K (we find the overfitting problem when finetuning GPT-2 on GSM8K).

## 5.4 ROBOT MANIPULATION

**Quantitative results.** We evaluate the proposed method of manipulating objects to achieve object relations specified by the textual descriptions (Text) or real-world images (Image). In Table 4, we find that using scorers of multiple camera views substantially improves the accuracy on both settings.

**Qualitative results.** Figure 4 shows the example results of the proposed method manipulating objects to accomplish the given task. Our method enables zero-shot robot manipulation on objects with different sizes, colors, and shapes given either the language goal or image goal.

Table 4: **Robot manipulation results on Ravens.** PIC can manipulate objects to achieve object relations specified by textual descriptions (Text) or real-world images (Image). Using scorers of multiple camera views substantially improves the success rate.

| Method Name | 2 Relations | | 3 Relations | |
|---|---|---|---|---|
| | Text ↑ | Image ↑ | Text ↑ | Image ↑ |
| **PIC (G+E1)** | 35.0 | 27.5 | 50.0 | 45.0 |
| **PIC (G+$\sum_{n=1}^{5}$ E$_n$)** | **67.5** | **52.6** | **75.0** | **65.3** |

## 6 ANALYSIS

PIC exhibits effective zero-shot generalization ability on a variety of tasks. To further understand the source of such generalization, we investigate two key components in PIC, *i.e.* the composition of multiple scorers (consensus optimization) (Section 6.1) and the iterative refinement (Section 6.2).

### 6.1 EFFECT OF CONSENSUS OPTIMIZATION

We have shown that composing multiple scorers contributes to zero-shot generalization. We further explore the influence of gradually adding each new scorer on the zeros-shot performance.

**Image generation.** In Table 5, we first show results of composing GLIDE and the CLIP scorer. We then gradually add a new scorer, the image classifier or classifier-free guidance, each time. Finally, we report the results of composing the generator and all scorers. The performance improves every time we add a new scorer, indicating that composing multiple scorers improves zero-shot performance.

Table 5: **Effect of composing multiple scorers.** Image generation results on ImageNet. Gradually adding new scorers keeps improving the performance.

| Method Name | Generator | Scorer | IS ↑ | FID ↓ | KID ↓ |
|---|---|---|---|---|---|
| **PIC (G+E1)** | GLIDE | CLIP | 25.017 | 30.462 | 6.174 |
| **PIC (G+E1+E2)** | GLIDE | CLIP + CLS | 30.438 | 29.543 | 5.435 |
| **PIC (G+E1+E3)** | GLIDE | CLIP + CLS-FREE | 30.500 | 29.726 | 4.304 |
| **PIC (G+E1+E2+E3)** | GLIDE | CLIP + CLS + CLS-FREE | **34.952** | **29.184** | **3.766** |

Table 6: **Effect of iterative refinement.** Grade school math results on GSM8K. PIC with iterative refinement outperforms baselines where the scorer only provides feedback to the generator at the end stage. BS is the beam search size.

| Method Name | Generator | Scorer | Interaction | BS=1 ↑ |
|---|---|---|---|---|
| **GPT-Pretrain+E** | GPT-2 (Medium) (Pretrain) | CLS | $t = T$ | 9.704 |
| **GPT-FT+E** | GPT-2 (Medium) (FT) | CLS | $t = T$ | 14.481 |
| **PIC (G+E)** | GPT-2 (Medium) (Pretrain) | CLS | $t = \{1, \cdots, T\}$ | **17.210** |

**Robot manipulation.** In Table 7, we analyze the effect of composing multiple scores on robot manipulation. The goal is specified by textual descriptions. Composing scores from multiple views, PIC (G+$\sum_{n=1}^{3} E_n$) and PIC (G+$\sum_{n=1}^{5} E_n$), leads to higher accuracy.

## 6.2 EFFECT OF ITERATIVE REFINEMENT

Next, we explore the influence of iterative refinement on zero-shot generalization, *i.e.* the feedback loop between the generator and scorers. We compare PIC with baselines that compose the generator and scorers, but with the scorers only providing feedback to the generator at the end.

**Grade school math.** In Table 6, the baselines, GPT-Pretrain+E and GPT-FT+E, generate five proposal solutions of a given math problem. Then the scorer, *i.e.* the same question-solution classifier used in PIC, selects the best solution based on its score. PIC iteratively refines the generated answer while the baselines refine the entirely generated solutions in the end. PIC and GPT-Pretrain+E use the same generator and scorer, but PIC outperforms GPT-Pretrain+E by %7.507. PIC still achieves better performance than GPT-FT+E, which uses a stronger generator (finetuned on the GSM8K dataset).

**Robot manipulation.** In Table 7, the baseline, No-IR (G+$\sum_{n=1}^{5} E_n$), first samples 100 trajectories without using the feedback from scorers. Then the scorers select the best trajectories based on the summed score. The generator and scorers of this baseline are the same as our method, *i.e.* PIC (G+$\sum_{n=1}^{5} E_n$), but our method outperforms the baseline by %37.5 on the "2 Relations" setting, indicating the effectiveness of iterative refinement in the proposed framework.

Table 7: **Effect of composing multiple scorers and iterative refinement on robot manipulation.** Both components are important for zero-shot generalization.

| Method Name | Interaction | 2 Relations | 3 Relations |
|---|---|---|---|
| **PIC (G+E1)** | $t = \{1, \cdots, T\}$ | 35.0 | 50.0 |
| **PIC (G+$\sum_{n=1}^{3} E_n$)** | $t = \{1, \cdots, T\}$ | 57.5 | 63.3 |
| **PIC (G+$\sum_{n=1}^{5} E_n$)** | $t = \{1, \cdots, T\}$ | **67.5** | **75.0** |
| **No-IR (G+$\sum_{n=1}^{5} E_n$)** | $t = T$ | 30.0 | 46.6 |

Together, these results show that the composition of multiple scorers and iterative refinement are both important for zero-shot generalization. These results point to the potential broader applicability of the proposed method as a general purpose framework for zero-shot multimodal tasks.

## 7 CONCLUSION AND FUTURE WORK

In this paper, we propose a unified framework for composing ensembles of pre-trained models through iterative consensus without any training or finetuning. Our framework consists of a generator and an ensemble of scorers. The scorers provide feedback to the generator to iteratively improve its generated results. We show the proposed method allows effective zero-shot generalization on four representative tasks, *i.e.* image generation, video question answering, grade school math, and robot manipulation, and even outperforms methods that directly finetune models on certain tasks. We further analyze the source of such zero-shot generalization by exploring the effect of the composition of multiple scorers and the iterative refinement, and find that both are important for zero-shot generalization.

As our method does not need any training or finetuning, one drawback is that its performance depends on the pre-trained models. Training large models are complementary to the framework and methods we proposed and may be directly applied. We hope to explore these directions for zero-shot generalization in future work. In addition, our framework enables the composition of separately trained models and boosts performance by leveraging the knowledge from multiple expert models. The scorers can be learned at different times on different data in an incremental-learning manner, enabling the combination of incrementally learned knowledge. Our framework thus paves the way for many potential applications in lifelong learning / continual learning settings.

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

# Appendix

In this appendix, we first show additional results in Appendix A. We then show experimental details of each task in Appendix B. The ethics statement of the Amazon Mechanical Turk experiment for video question answering is in Appendix C.

## A  ADDITIONAL RESULTS

### A.1  IMAGE GENERATION RESULTS

We show more images generation results using different scorer models in Fig. A1. We find that most of the time, the proposed framework, either using a single scorer model or composing multiple scorer models, work well as shown in the first four examples in the left column of Fig. A1. In some hard cases, some scorer models might fail (the rest examples in Fig. A1). However, there is no discernible trend on which scorer is better on what tasks. The results of composing multiple scorer models are significantly better than using a single one, as different scorer models capture different aspects of the information. For example, in the fifth example in the left column of Fig. A1, PIC with classifier-free guidance (CLS-FREE) and PIC with a pre-trained classifier (CLS) cannot generate an image with "tench", but PIC with the pre-trained CLIP (CLIP) can generate the correct result and the composed model (CLS-FREE + CLS + CLIP) also works. This is why we consider composing multiple scorers: to leverage the strength of each expert model and improve the worst case.

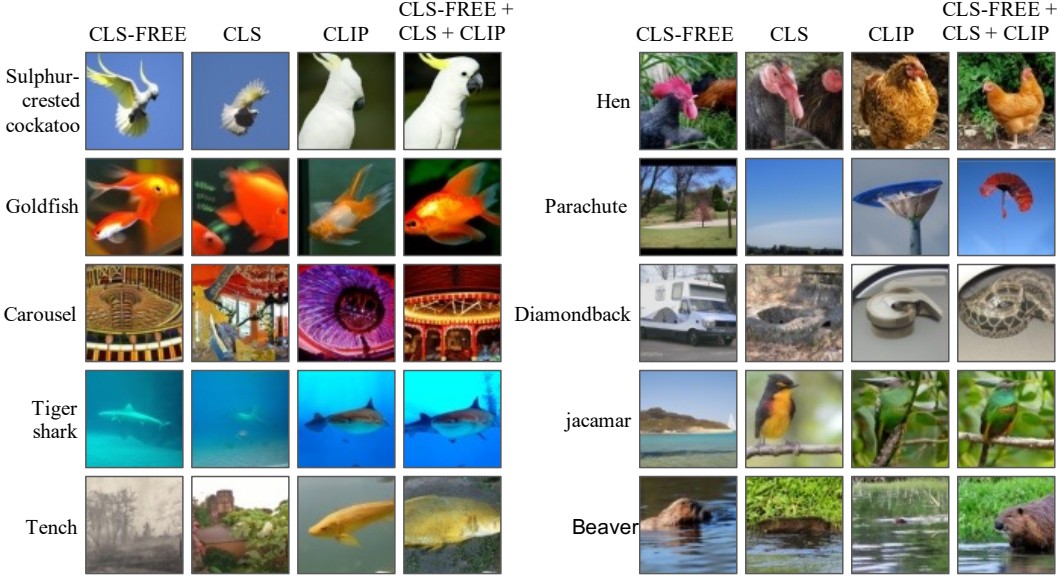

Figure A1: **Qualitative results.** Image generation results using different scorer models. Composing multiple scorers (CLS-FREE + CLS + CLIP) achieves the best performances.

### A.2  IMAGE GENERATION WITH DIFFERENT GENERATOR

Our framework can be applied to other generators as well. For example, we changed the generator, GLIDE, to Stable Diffusion (Rombach et al., 2021). The results of composing Stable Diffusion and classifier-free guidance (CLS-FREE) is shown in Table A1. Using a more powerful pre-trained model can further boost the performance.

### A.3  GRADE SCHOOL MATH QUALITATIVE RESULTS

Example results of different methods are shown in Fig. A2. Our method can solve math problems involving addition, subtraction, multiplication, and division, even for solutions with three-digit numbers. In contrast, GPT-FT often fails to understand math problems.

Table A1: **Our framework can be applied to other generators as well.** In the image generation task, we use a new generator, Stable-Diffusion (Rombach et al., 2021). Using a more powerful pre-trained model can further boost the performance. Image generation results on ImageNet are reported.

| Method Name | Generator | Scorer | IS ↑ | FID ↓ | KID ↓ |
|---|---|---|---|---|---|
| **PIC (G+E)** | GLIDE | CLS-FREE | 25.926 | 29.219 | 5.325 |
| **PIC (G+E)** | Stable-Diffusion | CLS-FREE | 31.689 | 29.546 | 6.562 |

| Grade school math questions | Ground Truth | GPT Pretrain | GPT FT | PIC (G+E) |
|---|---|---|---|---|
| Q: In a dance class of 20 students, 20% enrolled in contemporary dance, 25% of the remaining enrolled in jazz dance, and the rest enrolled in hip-hop dance. What percentage of the entire students enrolled in hip-hop dance? | 60 | A: 25% | A: 20 | A: 60 |
| Q: Melanie is a door-to-door saleswoman. She sold a third of her vacuum cleaners at the green house, 2 more to the red house, and half of what was left at the orange house. If Melanie has 5 vacuum cleaners left, how many did she start with? | 18 | A: 5 | A: 15 | A: 18 |
| Q: A fog bank rolls in from the ocean to cover a city. It takes 10 minutes to cover every 3 miles of the city. If the city is 42 miles across from the oceanfront to the opposite inland edge, how many minutes will it take for the fog bank to cover the whole city? | 140 | A: 10 | A: 10 | A: 140 |

Figure A2: **Grade school math example results.** Our method can solve math problems involving addition, subtraction, multiplication, and division.

## A.4 COMPOSING SCORER MODELS

The key idea of our method is to compose ensembles of pre-trained models, and the way to combine them can be variant. We add two additional experiments in this appendix: (1) using the best scorer (scorer that provides the highest score) and (2) using the weighted scores and add them together. Our method composes pre-trained models without training or finetuning, thus we did not learn separated weights for different models. But instead we add an experiment that uses the scorers (after softmax) of each scorer model as their weights and then composes the scorers using the weighted summed score. We compare these two baselines with our method that uses the summation of all scores. As shown in Table A2, using a summed score generates the best results.

Table A2: **Different ways to compose scorer models.** Composing scorers using their summed score generates the best results. Image generation results on ImageNet are reported.

| Method Name | Generator | Scorer | IS ↑ | FID ↓ | KID ↓ |
|---|---|---|---|---|---|
| **PIC (G+E1)** | GLIDE | CLIP | 25.017 | 30.462 | 6.174 |
| **PIC (G+E2)** | GLIDE | CLS | 22.077 | 30.871 | 7.952 |
| **PIC (G+E1+E2) Sum** | GLIDE | CLIP + CLS | **30.438** | **29.543** | **5.435** |
| **PIC (G+E1+E2) Max** | GLIDE | CLIP + CLS | 24.782 | 30.657 | 6.040 |
| **PIC (G+E1+E2) Weighted** | GLIDE | CLIP + CLS | 24.728 | 30.669 | 6.420 |

## A.5 ADDITIONAL BASELINE COMPARISONS

**Image Generation.** In Table A3, we compare our approach with a generative model specifically trained on ImageNet, Efficient-VDVAE (Hazami et al., 2022). Efficient-VDVAE is an unconditional hierarchical VAE model. As illustrated in Table A3, despite Efficient-VDVAE is explicitly trained on ImageNet, it is much worse than our approach.

**Robot Manipulation.** We compare our robot manipulation method with the robot manipulation model, CLIPort (Shridhar et al., 2022), used in the Socratic Models paper (Zeng et al., 2022). In Socratic models, the evaluation of robot manipulation consists of two steps: 1) a GPT model translates a text goal into a set of subgoals and 2) a CLIPort model executes each subgoal. In contrast, in our robot manipulation task, 1) a vision-language model is used to translate an image goal into a set of subgoals and 2) an iterative MPC procedure is used to execute each given subgoal. Thus a fair comparison is to compare the iterative MPC procedure used in our approach and the CLIPort model used in Socratic models. We thus compare our method to the multilingual CLIPort model on the performance of executing a single relation subgoal. CLIPort model obtains a success rate of 22.6% in this setting, while our approach obtains a success rate of 76.3%.

Table A3: **Comparison of our method and additional baselines.** Image generation results on ImageNet are reported. Our method outperforms the baseline.

| Method Name | Generator | Scorer | IS ↑ | FID ↓ | KID ↓ |
|---|---|---|---|---|---|
| **PIC (G+E1)** | GLIDE | CLIP | 25.017 | 30.462 | 6.174 |
| **PIC (G+E1+E2)** | GLIDE | CLIP + CLS | 30.438 | 29.543 | 5.435 |
| **PIC (G+E1+E3)** | GLIDE | CLIP + CLS-FREE | 30.500 | 29.726 | 4.304 |
| **PIC (G+E1+E2+E3)** | GLIDE | CLIP + CLS + CLS-FREE | **34.952** | **29.184** | **3.766** |
| **Efficient-VDVAE (Hazami et al., 2022)** | - | - | 6.566 | 143.642 | 23.007 |

## B  EXPERIMENTAL DETAILS

In this section, we provide more experimental details of each task. We use TITAN RTX 24GB GPUs for all the experiments.

### B.1  IMAGE GENERATION

We use the reverse diffusion process of GLIDE, a text-guided diffusion model, as the generator to generate image proposals. At each step of the diffusion process (corresponding to a step of the iterative refinement), we use the gradient from an ensemble of scorers to guide and update the generated proposals. We iteratively repeat this procedure until the final step.

As shown in Fig. A3, the image $x^k$ generated at iteration $k$ is first sent to the diffusion model to generate an image proposal $\hat{x}^{k+1}$. The scorers provide feedback to refine the generated result. The CLIP model computes the cosine similarity between the image and text features as the score (we used the pre-trained CLIP model from (Ho & Salimans, 2022).). The image classifier (Dhariwal & Nichol, 2021) predicts the probability of the image matching the text label as the score. The scores generated by different scorers are summed, and their gradient with respect to $x^k$ is used to compute the next reverse prediction $x^{k+1}$. The classifier-free guidance (Ho & Salimans, 2022) can be treated as an implicit classifier that directly provides pixel-wise gradient feedback to the generated image. Our framework enables the use of ensembles of different pre-trained models as scorers, significantly improving the zero-shot results by leveraging the strengths of multiple expert models.

Our implementation for image generation is modified based on the code of GLIDE (Nichol et al., 2021) and the classifier guidance diffusion (Dhariwal & Nichol, 2021). We use DDIM to sample images from GLIDE in 100 steps. The guidance scale is set to 3.

### B.2  VIDEO QUESTION ANSWERING

In video question answering, we use the proposed method to generate captions for the video frames and then use GPT-3 to summarize the captions to answer questions. We use GPT-2 as the generator and a set of CLIP models as scorers to generate captions for each video frame. The CLIP models (Radford et al., 2021; Reimers & Gurevych, 2019) are from the Huggingface library (Wolf et al., 2019):

- CLIP-32: `https://huggingface.co/openai/clip-vit-base-patch32`.

- CLIP-14: `https://huggingface.co/openai/clip-vit-large-patch14`.

- CLIP-multilingual: `https://huggingface.co/sentence-transformers/clip-ViT-B-32-multilingual-v1`.

Fig. A4 shows the framework for generating frame captions. Given a video frame $I$, we generate a sequence of words to describe it. To integrate feedback from scorers to the generator, similar to ZeroCap (Tewel et al., 2021), we define a context cache $C_t$ (a set of embedding functions in GPT-2, such as the embedding functions, $K$, $Q$, $V$, in the Transformer blocks.) that stores the context information generated so far, which is updated iteratively based on the feedback from scorers. The prediction of the next word from the generator $G$ is given by $x_{t+1} = G(x_t, C_t)$, where $G$ is the pre-trained language model. ZeroCap uses the following loss function to optimize $C_t$:

$$\arg\min_{C_t}\Big(\mathcal{L}_{\text{CLIP}}\left(\text{G}\left(x_t, C_t\right), I\right) + \lambda\mathcal{L}_{\text{CE}}\big(\text{G}\left(x_t, C_t\right), \hat{x}_{t+1}\big)\Big), \tag{A1}$$

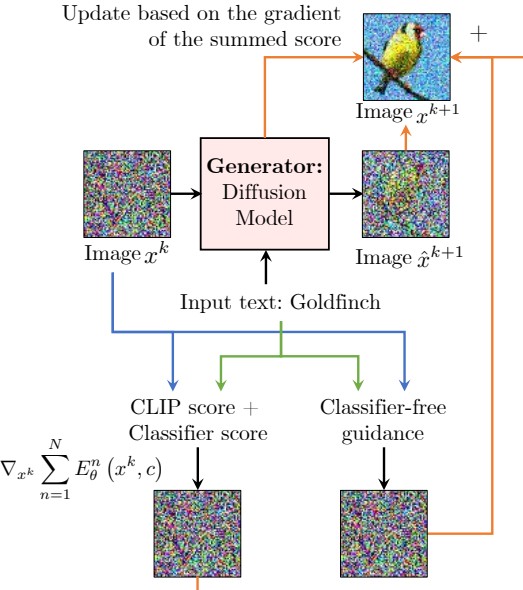

Figure A3: **Overview of image generation.** We use the reverse diffusion process of GLIDE (Nichol et al., 2021), a text-guided diffusion model, as the generator to generate image proposals. At each step of the diffusion process (corresponding to a step of the iterative refinement), we use the gradient from an ensemble of scorers, such as CLIP (Radford et al., 2021), to guide and update the generated proposals. The image $x^k$ generated at iteration $k$ is first sent to the diffusion model to generate an image proposal $\hat{x}^{k+1}$. The scorers provide feedback to refine the generated result. The CLIP model computes the cosine similarity between the image and text features as the score. The image classifier (Dhariwal & Nichol, 2021) predicts the probability of the image matching the text label as the score. The scores generated by different scorers are summed, and their gradient with respect to $x^k$ is used to compute the next reverse prediction $x^{k+1}$. Classifier-free guidance (Ho & Salimans, 2022) can be treated as an implicit classifier that directly provides pixel-wise gradient feedback to the generated image. We iteratively repeat this procedure until the final step. Our framework enables the use of ensembles of different pre-trained models as scorers, significantly improving the zero-shot results by leveraging the strengths of multiple expert models.

where $I$ is the feature of a video frame and $\hat{x}_{t+1}$ is the next word predicted by the original language model. The CLIP loss $\mathcal{L}_{\text{CLIP}}$ optimizes $C_t$ to make the new generated sentence describe the video frame. The second loss $\mathcal{L}_{\text{CE}}$ ensures the new generated sentence is close to the sentence generated by the original language model.

Our implementation is based on the code of ZeroCap (Tewel et al., 2021). The context cache $C_t$ is updated using:

$$C_t \longleftarrow C_t + \alpha \frac{\nabla_{C_t} p\left(x_{t+1} \mid C_t\right)}{\left\|\nabla_{C_t} p\left(x_{t+1} \mid C_t\right)\right\|^2}, \tag{A2}$$

where $p(x_{t+1}|C_t)$ is the probability of predicting word $x_{t+1}$ given $C_t$. Optimizing Eq. (A1) can be achieved by conducting the gradient descent using Eq. (A2). In our experiments, we use 5 steps of gradient descent. The learning rate $\alpha$ is set to 0.3.

In the video question answering tasks, we compose multiple CLIP scores and use their composed score to optimize $C_t$:

$$\arg \min_{C_t} \Big( \mathcal{L}_{\text{CLIP-32}} \left( G\left(x_t, C_t\right), I\right) + \mathcal{L}_{\text{CLIP-14}} \left( G\left(x_t, C_t\right), I\right) \tag{A3}$$

$$+ \mathcal{L}_{\text{CLIP-multilingual}} \left( G\left(x_t, C_t\right), I\right) + \lambda \mathcal{L}_{\text{CE}} \left( G\left(x_t, C_t\right), \hat{x}_{t+1}\right) \Big). \tag{A4}$$

After several iterations, the updated $C_t$ is used to generate the next token $x_{t+1} = G(x_t, C_t)$. We repeat this process until we generate the entire caption.

To answer the video questions, we cascade the generated captions of the video frames and the questions about this video to prompt GPT-3 to generate answers. For each video, we delete the first

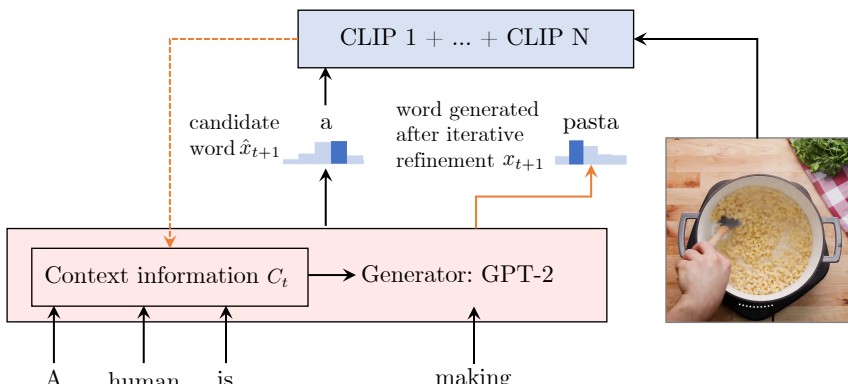

Figure A4: **Overview of video frame captioning for video question answering.** We use GPT-2 as the generator and a set of CLIP models as scorers to generate captions for each video frame. To integrate feedback from scorers to the generator, similar to ZeroCap (Tewel et al., 2021), we define a context cache $C_t$ (a set of embedding functions in GPT-2) that stores the context information generated so far, which is updated iteratively based on the feedback from scorers. To update $C_t$, we first use $G$ to generate a set of candidate words $\hat{X}_{t+1} = \{\hat{x}_{t+1}\}$, and then use the feature distance (after softmax) between each sentence (the concatenation of previous words and each new word $\{x_1, x_2, \cdots, \hat{x}_{t+1}\}$, where $\hat{x}_{t+1} \in \hat{X}_{t+1}$) and the video frame as the probability of them matching. The CLIP score is the cross-entropy loss $\mathcal{L}_{\text{CLIP}}$ between this new probability distribution and the original distribution of the next word obtained from the generator $G$ (see Equation 4 in (Tewel et al., 2021)). The gradient of summed scores (multiple CLIP models) is propagated to $G$ to update $C_t$ (see Equation 5 in (Tewel et al., 2021)). After several iterations, the updated $C_t$ is used to generate the next token $x_{t+1} = G(x_t, C_t)$. We repeat this process until we generate the entire caption. We cascade the captions of multiple video frames and questions about this video to prompt GPT-3 for video question answering.

# Q: how many people are there in the video
# A: 2
# Q: what is behind the person in white clothes
# A: tree
# Q: what is in front of the person with braid
# A: chair
...
# Q: what is the person in white doing
# A: tie hair
# Q: what happened to the person in gray after he threw a goal
# A: clap with your teammates
# Summarize the following descriptions and answer the question as shown above:
a Video showing the new Hair tutorial; a video showing young blond hair clip attaching to top pony tail of teens hair; ...; a video on the head hair clip website showing blonde long hair twisted in two knots.

# Q: is the person with a golden hair long hair

Figure A5: **Prompt given to GPT-3 for video question answering.** Text in black contains the question-answer pairs randomly sampled from the ActivityNet-QA training dataset. Text in blue has the video frame captions generated by the proposed method. Text in orange is the question about this video that needs to be answered.

10 frames and the last 10 frames to remove the beginning or ending advertisements. We then take 30 video frames evenly from the rest frames and send them to GPT-3. To guide GPT-3 to generate proper answers, we randomly select 30 question-answer pairs from the training set of ActivityNet-QA (Yu et al., 2019) and use them as part of the prompt of GPT-3. As shown in Fig. A5, the prompt of GPT-3 consists of examples of question-answer pairs, the video frame captions generated by the proposed method, and the question about this video that needs to be answered. The text generated by GPT-3 is used as the answer to the question asked. We also used the profanity check tool (`https://github.com/vzhou842/profanity-check`) to remove the improper answers.

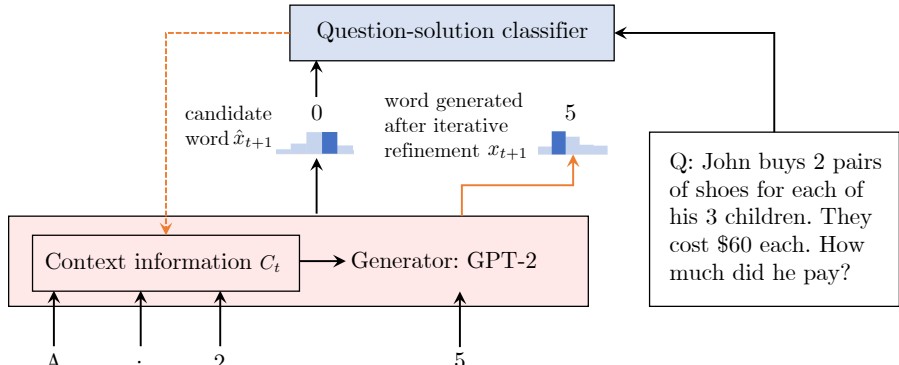

Figure A6: **Overview of solving grade school math problems.** We use GPT-2 as the generator and treat the grade school math problem as a text generation problem. The scorer, a pre-trained question-solution classifier, provides the generator feedback to guide the next token's generation $x_{t+1}$. We follow the approach used in VQA to iteratively optimize the generations based on the feedback from scorers. Our generator $G$ first generates a set of candidate words $\hat{X}_{t+1} = \{\hat{x}_{t+1}\}$, and then the classifier predicts the probability of each solution (the concatenation of previous words and each new word $\{x_1, x_2, \cdots, \hat{x}_{t+1}\}$, where $\hat{x}_{t+1} \in \hat{X}_{t+1}$) matching the given question. The classifier score is the cross-entropy loss between this new probability distribution and the original distribution of the next word obtained from the generator $G$. The gradient of the classifier score is used to update $C_t$ through iterative refinement (see Equation 5 in (Tewel et al., 2021)). The updated $C_t$ is used to predict the next word $x_{t+1} = G(x_t, C_t)$. We repeat this process until we generate the complete solution.

### B.3 GRADE SCHOOL MATH

We treat the grade school math problem as a text generation problem. As shown in Fig. A6, we use GPT-2 as the generator and a pre-trained question-solution classifier as the scorer. The pre-trained classifier is a binary classifier trained on the training set of GSM8K (Cobbe et al., 2021). Given a math problem, such as "Natalia sold clips to 48 of her friends in April, and then she sold half as many clips in May. How many clips did Natalia sell altogether in April and May?", and an answer, such as "72". If the answer is correct for the given problem, then the label is 1; otherwise, the label is 0.

After training, the classifier is used as the scorer to provide feedback to the generator to guide the next token's generation $x_{t+1}$. Similar to VQA, the generator $G$ first generates a set of candidate words $\hat{X}_{t+1} = \{\hat{x}_{t+1}\}$, and then the classifier predicts the probability of each solution (the concatenation of previous words and each new word $\{x_1, x_2, \cdots, \hat{x}_{t+1}\}$, where $\hat{x}_{t+1} \in \hat{X}_{t+1}$) matching the given question. The classifier score is the cross-entropy loss between this new probability distribution and the original distribution of the next word obtained from the generator $G$ (the way to compute the classifier score is the same as computing the CLIP score in VQA). We also used the cross-entropy loss $\mathcal{L}_{CE}$ in Equation 2 of ZeroCap (Tewel et al., 2021) to ensure the generated sentence is grammatically sound. The context cache $C_t$ is updated in the same way as the video question answering task, but we use the classifier score when providing the feedback to $C_t$. The updated $C_t$ is used to predict the next word $x_{t+1} = G(x_t, C_t)$. We repeat this process until we generate the complete solution. Similarly to the video question task, we use 5 steps of gradient descent. The learning rate $\alpha$ is set to 0.3.

### B.4 ROBOT MANIPULATION

In robot manipulation, we use the proposed method to manipulate objects in Ravens (Zeng et al., 2020) to conform to a set of object relations specified by text descriptions or real-world images. We use MPC+World Model as the generator and ViLD (Gu et al., 2021) as the scorer. As shown in Figure A7, given a real-world image, our model manipulates objects in the environment to achieve a state with objects having the same object relations as the given image. We first use ViLD to generate a 2D segmentation of the real-world image and the corresponding text label, such as "mug", for each segment. We then use the relative pixel-wise offsets of segmentation masks and the text labels to infer a set of object relations (top panel of Figure A7).

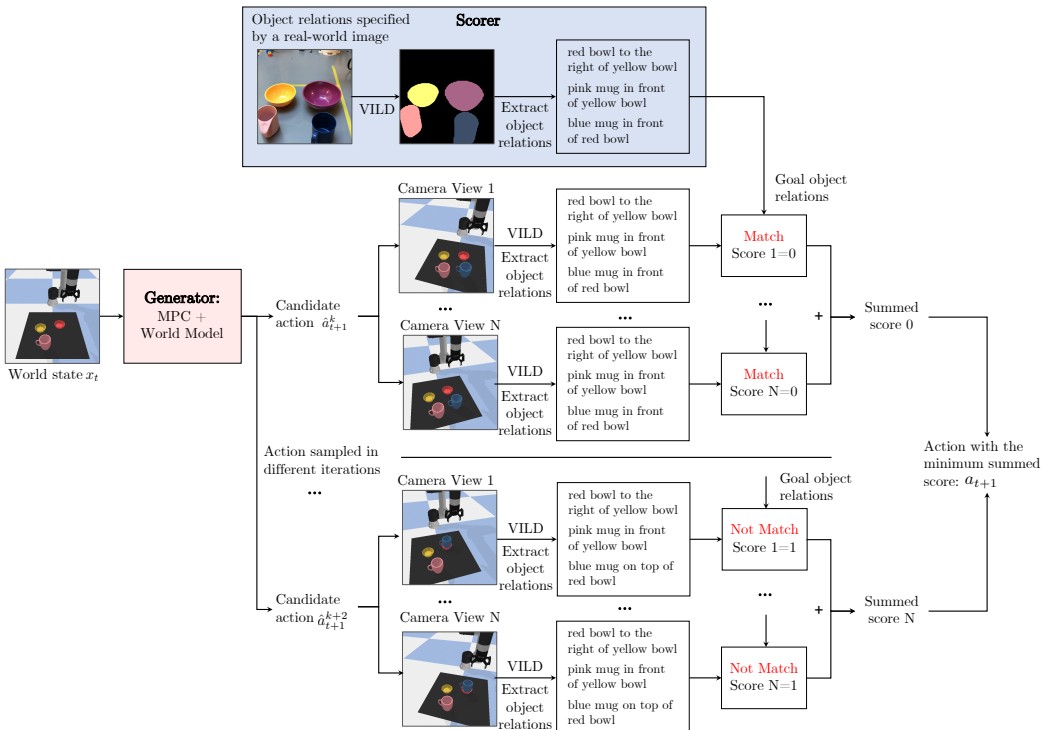

Figure A7: **Overview of robot manipulation.** We use MPC+World Model as the generator and ViLD as the scorer to manipulate objects to conform to a set of object relations specified by text descriptions or real-world images. **Top:** given a real-world image, we first use ViLD to generate a 2D segmentation of the real-world image and the corresponding text label, such as "mug", for each segment. We then use the relative pixel-wise offsets of segmentation masks and the text labels to infer a set of object relations. **Bottom:** Given the current world state $x_t$, we aim to generate an action $a_{t+1}$ so that the new world state after executing $a_{t+1}$ has object relations closer to the object relations in the given image. To do this, we first use the generator (MPC+World model) to generate a set of candidate actions $\{\hat{a}_{t+1}^k\}$ and the corresponding world states $\{\hat{x}_{t+1}^k\}$ after executing each candidate action. For each new world state $\hat{x}_{t+1}^k$, we render $N$ 2D images from $N$ camera views. Each rendered image is sent to VILD to get a segmentation map and text labels. We project the objects into 3D space based on the segmentation map and the depth map of the image. We then obtain the object relations based on their 3D positions and predicted text labels. We compare the object relations obtained from each rendered image and the object relations obtained from the real-world image to compute the score. The score is 0 if the relations are matching; otherwise, 1. We sum the scores from each rendered image to obtain the final score. We choose the action $a_{t+1}$ that leads to a world state with the minimum summed score. We execute $a_{t+1}$ in the environment and get a new state $x_{t+1}$. We repeat this process until the task is accomplished or we are at the final step $T$.

Given the current world state $x_t$, we aim to generate an action $a_{t+1}$ so that the new world state after executing $a_{t+1}$ has object relations closer to the object relations in the given image. To do this, we first use the generator (MPC+World Model) to generate a set of candidate actions $\{\hat{a}_{t+1}^k\}$ and the corresponding world states $\{\hat{x}_{t+1}^k\}$ after executing each candidate action. For each new world state $\hat{x}_{t+1}^k$, we render $N$ 2D images from $N$ camera views. Each rendered image is sent to VILD to get a segmentation map and text labels. We project the objects into 3D space based on the segmentation map and the depth map of the image. We then obtain the object relations based on their 3D positions and the predicted text labels. We compare the object relations obtained from each rendered image and the object relations obtained from the real-world image to compute the score. The score is 0 if the relations are matching; otherwise, 1. We sum the scores from each rendered image to obtain the final score. We choose the action $a_{t+1}$ that leads to a world state with the minimum summed score. We execute $a_{t+1}$ in the environment and get a new state $x_{t+1}$. We repeat this process until the task is accomplished or we are at the final step $T$, where $T$ equals to the number of relations extracted from the real-world image.

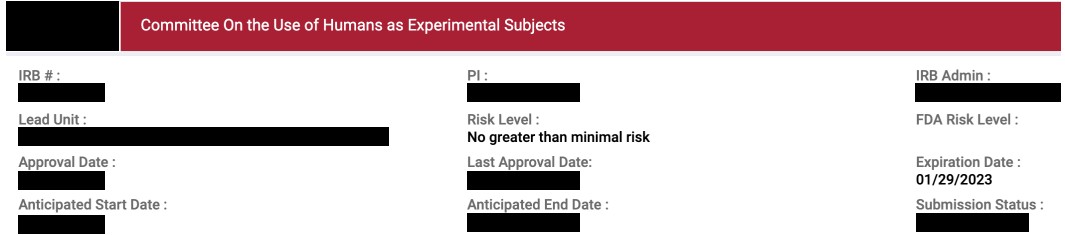

Figure A8: Screenshot of the approval form from the Committee on the Use of Humans as Experimental Subjects.

### B.5 A UNIFIED FRAMEWORK FOR COMPOSING PRE-TRAINED MODELS

Our method shares some similar architecture with existing works, such as ZeroCap (Tewel et al., 2021) and CLIP-guided diffusion models (Nichol et al., 2021). However, the focus of our paper is to propose a general framework for composing different pre-trained models across a variety of tasks, and these particular methods are concrete instantiations of our proposed framework. In addition, in this work, we also illustrate how we may combine ensembles of different pre-trained models as scorers to leverage the "wisdom of the crowds" where each scorer provides complementary feedback to the generator, compensating for the potential weaknesses of other scorers. Through iterative optimization and the composition of multiple scorers, our method shows effective zero-shot generalization ability on various multimodal tasks.

## C  ETHICS STATEMENT OF AMAZON MECHANICAL TURK EXPERIMENTS

To evaluate approaches on solving the zero-shot video question answering tasks, we ask workers from Amazon Mechanical Turk to evaluate the generated answer based on the video and the asked question. Before showing the questions and answers to the workers, we used the profanity check tool (`https://github.com/vzhou842/profanity-check`) to remove the improper questions and answers. As shown in Fig. A8, this experiment was approved by the Committee on the Use of Humans as Experimental Subjects. A screenshot of the task is shown in Fig. A9. The instructions shown to participants are listed as follows:

> **Instructions:** By making judgments about these questions and answers, you are participating in a study being performed by [XXX]. Your participation in this research is voluntary. You may decline further participation, at any time, without adverse consequences. Your anonymity is assured; the researchers who have requested your participation will not receive any personal information about you.

Given a video, a question, and a generated answer, the workers from Amazon Mechanical Turk measure whether the answer is correct for the given question and video. Each video shows three question-answer pairs (only one question-answer pair is shown in the screenshot). The answers are generated by different methods. The workers are not told which method generates each answer. The workers are asked to choose "yes" or "no". If the worker thinks the answer matches the given video and question, they should choose "yes"; otherwise, "no".

To control the quality, each task is evaluated by three different workers. The workers are required to have an approval rate greater than $98\%$. Our test shows that each task takes around 10 seconds, but the workers are given up to one hour to complete each task. The workers are paid $\$0.05$ for finishing each task with an estimated hourly payment of $\$18$, more than the United States federal minimum wage. There are 33 workers in total who joined our experiment.

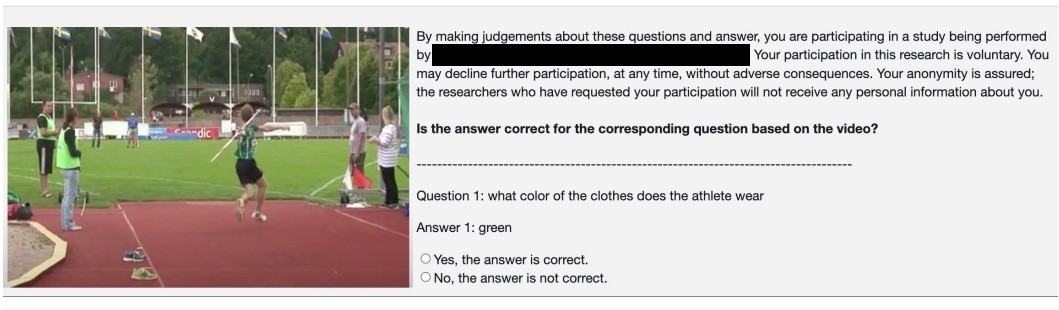

Figure A9: **Screenshot of Amazon Mechanical Turk we used for the video question answering experiment.** Workers are shown a video, three questions, and the answer to each question. The answers are generated by different methods. The workers are not told which method generates each answer. The workers are asked to select "yes" or "no" based on their measurement of whether the answer is correct for the given video and question.

