# OpenReview forum: "Composing Ensembles of Pre-trained Models via Iterative Consensus"
_ICLR.cc/2023/Conference — ICLR 2023 poster_

### Official Review · Reviewer_BdG8 · 2022-10-24

**Confidence:** 4
**Correctness:** 4
**Technical Novelty And Significance:** 3
**Empirical Novelty And Significance:** 3
**Recommendation:** 8

**Clarity, Quality, Novelty And Reproducibility:**

The text is quite clear. The experiments have been defined explicitly in the Appendix. The presentation of the ideas in the text can be more coherent but still sets a high bar of excellence.

The technique is novel, it provides a simple yet effective strategy of combining the strengths of related but relatively orthogonal models. The proposition of using a class of models as diverse as GLIDE, VilD, CLIP in the same framework is novel.

The technique proposed in the paper is quite clearly described and should be straight forward to reproduce it.

**Details Of Ethics Concerns:**

No ethical concerns.

**Strength And Weaknesses:**

Strengths:

* Elegant technique for combining the strengths of complementary models which are trained on large scale dataset
* No retraining required which is expensive in most cases with large scale models
* Results show effectiveness of the technique

Weaknesses:

* The change in the scores with the proposed technique seems minimal.
* In industry use cases the GPU memory would overflow to fit all the models simultaneously.
* The greatest benefit of this technique seems to be for Robotic Manipulation, reducing the scope of impact.

**Summary Of The Paper:**

This paper proposes an ensembling framework for use the capabilities of multiple foundational models. In cases where there isn't a unified interface to interact with a number of foundational models, it provides a unified framework to allow multiple foundational models to interact with each other. The paper proposes to divide the foundational models as "Generators" and "Evaluators". The "Generator" set of models generates proposals and the "Evaluator" sets of models evaluates the proposals and are used to iteratively refine the proposal using its gradients.

**Summary Of The Review:**

This paper proposes a simple technique for combining the strengths of several large scale foundational models. It is especially interesting when large scale data across multiple domains is not readily available such as Robotics and Language Models. It would have been interesting to see some tasks which are only enabled with this system which would not be possible without the presence of this system.

---

> ### Author Response · Authors · 2022-11-18
> **Reviewer BdG8 Response**
>
> Dear Reviewer, thank you for your valuable comments and feedback. We appreciate that you think our framework is elegant and the results are impressive. We have addressed your questions below. Please let us know if you have any additional questions -- we are happy to clarify or provide additional experiments.
>
> **Q1) The change in the scores with the proposed technique seems minimal.**
>
> Thanks for your questions. We have provided detailed explanations to your question in the “general response to all reviewers” part (Q1, Q3).
>
> **Q2) In industry use cases the GPU memory would overflow to fit all the models simultaneously.**
>
> Thanks for your questions. We have provided detailed explanations to your question in the “general response to all reviewers” part (Q4).
>
> **Q3) The greatest benefit of this technique seems to be for Robotic Manipulation, reducing the scope of impact.**
>
> Thanks a lot for your feedback. We think this is a general framework that can be applied to solve different tasks. We believe other tasks, such as image generation, video question answering, and solving math problems are also important research problems with many downstream applications. Furthermore, with the advent of increasingly large and diverse pre-trained models, this framework can be directly applied to many other new tasks as well.

---

### Official Review · Reviewer_To1A · 2022-10-24

**Confidence:** 3
**Clarity, Quality, Novelty And Reproducibility:** Approach looks very novel and high qu…
**Correctness:** 4
**Technical Novelty And Significance:** 4
**Empirical Novelty And Significance:** 4
**Recommendation:** 8

**Strength And Weaknesses:**

Strengths:
- The framework is very elegant. It uses the complimentary strengths of, for example, language models (as generators) and retrieval models like CLIP (as scorers).
- The notion of iterative feedback to get consistency between the generator's output and the scorer is innovative.
- The results, especially the zero-shot results, are quite impressive.

Weaknesses
- This proposed framework is very similar to the kind of optimization that appears in older literature on MAP estimation in structured prediction. A discussion on this would be useful.
- Also worth discussing would be the effect of the limitations of the pretrained models on the proposed framework.


**Summary Of The Paper:**

This paper proposes a new framework for using pre-trained models for various recognition and reasoning tasks. The idea is to use a "generator" and an ensemble of "scorers" to iteratively improve the output: the generator proposes an output that the scorer scores. This general framework is demonstrated on multiple tasks, including visual question answering, solving math problems, image generation and robot manipulation.

**Summary Of The Review:**

I find the general framework quite interesting and novel. I think the connections to traditional iterative methods for structured prediction problems are very intriguing and should be discussed. That said, I find the approach innovative enough to be published.

---

> ### Author Response · Authors · 2022-11-18
> **Reviewer To1A Response**
>
> Dear Reviewer, thank you for your valuable comments and feedback. We appreciate that you think our framework is very elegant and the results are quite impressive. We have addressed your questions below. Please let us know if you have any additional questions -- we are happy to clarify or provide additional experiments.
>
> **Q1) This proposed framework is very similar to the kind of optimization that appears in older literature on MAP estimation in structured prediction. A discussion on this would be useful.**
>
> We have added a discussion to Structure Prediction Energy Networks [5] in the related work of our paper – it uses a similar idea of using energy optimization to refine the final structural prediction. Feel free to let us know any other additional references we should add to the paper.
>
> [5] David Belanger, Bishan Yang, Andrew McCallum. End-to-End Learning for Structured Prediction Energy Networks
>
> **Q2) Also worth discussing would be the effect of the limitations of the pretrained models on the proposed framework.**
>
> Thanks a lot for your suggestion. The limitation of using pretrained models is in Section 7 second paragraph.

---

### Official Review · Reviewer_dTDv · 2022-10-27

**Confidence:** 4
**Correctness:** 4
**Technical Novelty And Significance:** 3
**Empirical Novelty And Significance:** 2
**Recommendation:** 6

**Clarity, Quality, Novelty And Reproducibility:**

The quality and clarity of the work is good with the problem statement properly conveyed to the reader. The introduction is quite comprehensive covering the current state of research in the topic. Related works are well discussed. The method used for solving the four zero-shot tasks have been properly explained and are novel. But as mentioned in the weaknesses section above, extensive experiments and analysis is missing, which is very important for understanding the advantage of the proposed framework.

The use of Amazon Mechanical Turk to get the accuracy for the video question answering task hinders easy reproducibility of the corresponding experiments. An alternate metric not involving the requirement of human feedback would be encouraged to make the experiments more reproducible. Additionally, including details of some hyperparameters (check weaknesses) can increase reproducibility.


**Strength And Weaknesses:**

**Strengths**:

- The paper writing, presentation, and figures are very good and well-structured.
- The motivation to solve the problem of composing pretrained models is interesting and differences with prior works has been well highlighted.

**Weaknesses**:

-  As the main motivation of the paper is to leverage the individual expertise of the pretrained models, it would be great to have some qualitative analysis on the same. Specifically, as can be seen from Table 1 that the best performance is obtained when all the scorers E1, E2, E3 are used as an ensemble, more insights on identifying individual expertise of the scorers can make the submission stronger. E.g. “which groups of classes does E1 expertise which E2 and E3 fails to?”, “What information does CLS-FREE provide which CLIP and CLS do not?”, “How do the results change with different generators?”. Getting answers to similar questions would be very interesting.

- How about selecting one best scorer for providing feedback to the generative model? I understand that in the context of 3 scores it does not make much sense and the results already show that the combination works the best. However, in many practical scenarios given that HuggingFace models are easily available, there will be access to 100s of diverse models that can be considered as scorers. In those scenarios, is it still advisable to consider the combination for providing feedback to the generator? Authors should perform experiments with a diverse set of models trained using different datasets and of different sizes to verify how the capacity and knowledge of the individual models as scorer effects the final performance. The experiments in the current form are limited to demonstrate the potential of the method.

- What about confidence of predictions for the different scorers? Ideally there should be a learnable weight signifying the importance of a scorer for a particular task and sample. I would encourage the authors to perform experiments for verifying the effect of such weights in the final performance.

- Most of the experiments mainly compare the performance of PIC with different variants of the same method. Given that the proposed framework focuses on composing different models, how is comparable to state-of-the-art methods, e.g, Image generation or video captioning? Due to the lack of comparison with SOTA methods, it is not clear why one should use this framework in their application instead of using any other existing methods.

- What about the memory requirement and FLOPs during inference and training of this method comparable to existing baselines?

- How were the accuracies for the compared baselines JustAsk and JustAsk (FT) computed in Table 2? Was Amazon Mechanical Turk used for them as well?

- It would be great to have a baseline experiment in which PIC is asked to select an answer from the same pre-defined set used for other baselines instead of using a free-form response approach. How is that baseline compared to state-of-the-art methods on the same dataset? A thorough comparison and analysis should be performed to justify the advantage and utility of the proposed framework.

- As mentioned for Image Generation, analyzing the contribution of each of the scorers would be useful. E.g. “What attributes does CLIP-multilingual specialize in?” Analysis on the usage of GPT-2 is missing. How much does GPT-2 contribute towards the overall performance in video question answering?

- The paper also lacks some implementation details of the optimization procedure. Specifically, details like number of iterative steps used for optimizing for different tasks, learning curves with iterations (can be included in supplementary), optimization time required for each task, details of the context cache - size, initialization, etc.

- Experiments showing applicability of the framework to different permutations of possible generators and scorers are missing. E.g. getting results by using different generators for a given task.

- Experimental comparison with the discussed prior work Socratic Models (Zeng et al., 2022) is also not provided. If it is not possible to compare them in the experimented tasks due to language interface, how about comparing the proposed framework on the tasks used for zero-shot evaluation on Socratic Models.

**Summary Of The Paper:**

This paper focuses on the problem of leveraging individual expertise of various large pretrained models to compose an ensemble for solving various multi-modal problems in a zero-shot manner. The authors propose a unified framework, called PIC, in which the pretrained models are composed using closed-loop iterative consensus optimization. Specifically, for a given task, the framework employs a single model as a generator and an ensemble of scorers. The generator iteratively generates proposals and each scorer provides a feedback score indicating their agreement, then the generator refines its output until all the scorers achieve a final consensus. Experiments are shown on four zero-shot tasks - image generation, video question answering, grade school math, and robot manipulation. Results show that a composition of the pretrained models with an ensemble of scorers outperform baselines with a single scorer by a considerable margin.

**Summary Of The Review:**

I think the identified problem is important but the proposed method is a bit naive and experiments are not convincing. The proposed method should be evaluated and analyzed using extensive experiments resonating with the motivation to solve the problem. In the rebuttal I expect new experiments and analysis listed in the weaknesses to convince me that the proposed method has practical use and outperforms the missing baselines and exiting methods.

---

> ### Author Response · Authors · 2022-11-18
> **Reviewer dTDv Response (1/2)**
>
> Dear Reviewer, thank you for your valuable comments and feedback. We appreciate that you think our motivation to solve the problem of composing pretrained models is interesting and differences with prior works has been well highlighted. We have addressed your questions below. Please let us know if you have any additional questions -- we are happy to clarify or provide additional experiments.
>
> **Q1) As the main motivation of the paper is to leverage the individual expertise of the pretrained models, it would be great to have some qualitative analysis on the same. Specifically, as can be seen from Table 1 that the best performance is obtained when all the scorers E1, E2, E3 are used as an ensemble, more insights on identifying individual expertise of the scorers can make the submission stronger. E.g. “which groups of classes does E1 expertise which E2 and E3 fails to?”, “What information does CLS-FREE provide which CLIP and CLS do not?”, “How do the results change with different generators?”. Getting answers to similar questions would be very interesting.**
>
> Thanks a lot for your insightful suggestion. We have added images generation results using different scorer models in Appendix C.1. We find that most of the time, the proposed framework, either using a single scorer model or composing multiple scorer models, work well as shown in the first four examples in the left column of Appendix Fig. A8. In some hard cases, some scorer models might fail (the rest examples in Appendix Fig. A8). However, there is no discernible trend on which scorer is better on what tasks. The results of composing multiple scorer models are significantly better than using a single one, as different scorer models capture different aspects of the information.
>
> For example, in the fifth example in the left column of Appendix Fig. A8, PIC with classifier-free guidance (CLS-FREE) and PIC with a pre-trained classifier (CLS) cannot generate an image with “tench”, but PIC with the pre-trained CLIP (CLIP) can generate the correct result and the composed model (CLS-FREE + CLS + CLIP) also works. This is why we consider composing multiple scorers: to leverage the strength of each expert model and improve the worst case.
>
> **Q2) How about selecting one best scorer for providing feedback to the generative model? I understand that in the context of 3 scores it does not make much sense and the results already show that the combination works the best. However, in many practical scenarios given that HuggingFace models are easily available, there will be access to 100s of diverse models that can be considered as scorers. In those scenarios, is it still advisable to consider the combination for providing feedback to the generator? Authors should perform experiments with a diverse set of models trained using different datasets and of different sizes to verify how the capacity and knowledge of the individual models as scorer effects the final performance. The experiments in the current form are limited to demonstrate the potential of the method.**
>
> Thanks a lot for your insightful questions. We have added an experiment that uses the best scorer in the “general response to all reviewers” part (Q3). In this paper, we provide results illustrating the benefits of composing pre-trained models across different tasks and demonstrate the effectiveness of using multiple sets of models on each task. In future work, we would be happy to extend our framework to compose more types of models, such as N=100. However, we believe the existing results already illustrate the benefit of our model composition.
>
> **Q3) What about confidence of predictions for the different scorers? Ideally there should be a learnable weight signifying the importance of a scorer for a particular task and sample. I would encourage the authors to perform experiments for verifying the effect of such weights in the final performance.**
>
> Thanks for your suggestion. We have provided an additional experiment that composes different scorers based on their weights in the “general response to all reviewers” part (Q3).
>
> **Q4) Most of the experiments mainly compare the performance of PIC with different variants of the same method. Given that the proposed framework focuses on composing different models, how is comparable to state-of-the-art methods, e.g, Image generation or video captioning? Due to the lack of comparison with SOTA methods, it is not clear why one should use this framework in their application instead of using any other existing methods.**
>
> Thanks a lot for your insightful suggestion. We have addressed your question about “lack of comparison with previous works” in the “general response to all reviewers” part (Q2).

---

> > ### Author Response · Authors · 2022-11-18
> > **Reviewer dTDv Response (2/2)**
> >
> > **Q5) How were the accuracies for the compared baselines JustAsk and JustAsk (FT) computed in Table 2? Was Amazon Mechanical Turk used for them as well?**
> >
> > Yes, they are evaluated by Amazon Mechanical Turk workers. Given a video, a question, and an answer generated by a method, the workers are asked to decide whether the answer is correct for the question and video.
> >
> > **Q6)  It would be great to have a baseline experiment in which PIC is asked to select an answer from the same pre-defined set used for other baselines instead of using a free-form response approach. How is that baseline compared to state-of-the-art methods on the same dataset? A thorough comparison and analysis should be performed to justify the advantage and utility of the proposed framework.**
> >
> > Thanks a lot for your suggestion. Existing methods treating the video question answering task as a classification problem have major limitations. For example, some questions may have multiple answers. Thus we select to do open-ended video question answering, which is more challenging but closer to real-world scenarios.
> > We are happy to add an experiment discussing this in the final version of this paper if the reviewer can suggest a suitable experimental setting to test.
> >
> > **Q7) As mentioned for Image Generation, analyzing the contribution of each of the scorers would be useful. E.g. “What attributes does CLIP-multilingual specialize in?” Analysis on the usage of GPT-2 is missing. How much does GPT-2 contribute towards the overall performance in video question answering?**
> >
> > Thanks a lot for your suggestion. We have shown the qualitative results in Appendix Fig. A8. We find that most of the time, the proposed framework, either using a single scorer model or composing multiple scorer models, work well as shown in the first four examples in the left column of Appendix Fig. A8. In some hard cases, some scorer models might fail (the rest examples in Appendix Fig. A8). However, there is no discernible trend on which scorer is better on what tasks. The results of composing multiple scorer models are significantly better than using a single one, as different scorer models capture different aspects of the information.
> >
> > GPT-2 is used to generate text, so it is not possible to do video question answering without using GPT-2. We are happy to add additional analysis in the paper if the reviewer can suggest a suitable experimental setting to test.
> >
> > **Q8) The paper also lacks some implementation details of the optimization procedure. Specifically, details like number of iterative steps used for optimizing for different tasks, learning curves with iterations (can be included in supplementary), optimization time required for each task, details of the context cache - size, initialization, etc.**
> >
> > Thanks for your suggestion. Our method does not require any training, thus we do have learning curves with iterations and optimization time. In this rebuttal, we have added more implementation inference details in Appendix A.
> >
> > **Q9) Experiments showing applicability of the framework to different permutations of possible generators and scorers are missing. E.g. getting results by using different generators for a given task.**
> >
> > Thanks for your suggestions. Our framework can be applied to other generators as well. For example, we changed the generator, GLIDE, to Stable Diffusion. The results of composing Stable Diffusion and classifier-free guidance (CLS-FREE) are IS: 31.689, FID: 29.546, and KID 6.525. Using a more powerful generator can future improve the performance. We have added the results in Appendix C.2.
> >
> > **Q10) Experimental comparison with the discussed prior work Socratic Models (Zeng et al., 2022) is also not provided. If it is not possible to compare them in the experimented tasks due to language interface, how about comparing the proposed framework on the tasks used for zero-shot evaluation on Socratic Models.**
> >
> > Thanks for your questions. We cannot directly compare our model and Socratic Models because the underlying interface through which we execute each task is different and because there is no well-defined performance metric on most of the evaluated tasks. In this rebuttal, we have added another baseline, CLIPort, which is used by Socratic Models for robot manipulation tasks. We have provided the results of CLIPort in the “general response to all reviewers” part (Q2) and have discussed why it is more natural to compare with CLIPport.

---

> > > ### Comment · Reviewer_dTDv · 2022-12-01
> > > **Final Response**
> > >
> > > I thank the authors for their response and effort on the new experiments. While authors response clarified most of my concerns, I am still not fully convinced by the comparison with state-of-the-art methods. While authors have added comparison with 2 additional methods during the rebuttal period, I think more comparisons and analysis on SOTA methods should be included in the paper. However, as the problem of composing pretrained models is interesting and the paper tries to propose a simple solution with promising performance, I am increasing my initial score to 6.

---

### Official Review · Reviewer_kbow · 2022-11-01

**Confidence:** 3
**Correctness:** 4
**Technical Novelty And Significance:** 2
**Empirical Novelty And Significance:** 3
**Recommendation:** 5

**Clarity, Quality, Novelty And Reproducibility:**

The clarity of some methods part is questionable, it is not friendly to readers. This paper has certain novelty in unification but the core idea is too similar to previous works. The reproducibility and quality seem okay to me.

**Strength And Weaknesses:**

Strengths:
1. This framework doesn't need further task-specific pre-training and finetuning. It only needs to iteratively update the pre-trained models to refine the generation result.
2. The idea of using scorers to guide generators to reach a consensus is interesting and seems to be generalizable to four different tasks as in this paper.

Weaknesses:
1. For the methodology, this paper mainly instantiate the proposed method into two procedures: (1) continuous updating on image generation. (2) Discrete optimization over samples.  As for the first one, the whole method is quite similar to VQGAN-CLIP[1] in that they both optimize the image generator's input by a CLIP model to generate a better image. The difference is: at VQGAN-CLIP, the image generator is VQGAN and the optimized input is a latent feature, while this work used a diffusion model and the optimized input is a noisy image. Despiting the minor difference, those two share similar intuition and working mechanisms. For the second procedure, the caption generation optimization is almost the same as ZeroCap[2]. As mentioned in Figure A4's caption, this paper exactly used Equation 5 in ZeroCap to do the optimization. So is the Grade School Math text generation task. Overall, Although I appreciate the author(s)' efforts in unifying the different tasks in the same framework of iterative optimization, I still think the distinction between previous works and methodology contributions to the community is limited.
2. For the experiments, all those four tasks lack enough comparison with previous works. In Zero-shot videoQA, only JustASK is compared. In GSM8K, only GPT is compared. Moreover, the comparison seems not fair to me because the proposed PIC assumes to have a question-solution classifier, while GPT-FT(Pretrain) does not. In image generation and robot manipulation, no other baseline methods are compared.
3. Possibly Wrong/Confusing Illustrations. a). In Figure 2c, how does the input video frame feed into GPT-2? b) The explanation of $L_{CLIP}$ in Video question answering (VQA) of Sec3.1 is so confusing. I cannot really understand until I go back to read ZeroCap.

Ref:
[1] VQGAN-CLIP: Open Domain Image Generation and Editing with Natural Language Guidance, ECCV2022.
[2] ZeroCap: Zero-Shot Image-to-Text Generation for Visual-Semantic Arithmetic, CVPR2022.

**Summary Of The Paper:**

This paper proposed to iteratively optimize a system of one generator and multiple classifiers(scorers) for zero-shot generation tasks. It unified four multimodal generation tasks under the same framework. And experiments show that it can have a satisfactory generation ability on those tasks.

**Summary Of The Review:**

Overall, I prefer to weak rejection because of limited novelty and lack of comparison.

---

> ### Author Response · Authors · 2022-11-18
> **Reviewer Kbow Response**
>
> Dear Reviewer, thank you for your valuable comments and feedback. We appreciate that you think the idea of using scorers to guide generators to reach a consensus is interesting and can be generalizable to different tasks. We have addressed your questions below. We also have added a baseline for both the image generation and robot manipulation tasks. Please let us know if you have any additional questions -- we are happy to clarify or provide additional experiments.
>
> **Q1) Difference from previous methods**
>
> Thanks for your questions. We have provided detailed explanations to your question in the “general response to all reviewers” part (Q1).
>
> **Q2) For the experiments, all those four tasks lack enough comparison with previous works. In Zero-shot videoQA, only JustASK is compared. In GSM8K, only GPT is compared. Moreover, the comparison seems not fair to me because the proposed PIC assumes to have a question-solution classifier, while GPT-FT(Pretrain) does not. In image generation and robot manipulation, no other baseline methods are compared.**
>
> Thanks a lot for your insightful questions. We have addressed your question about “lack of comparison with previous works” in the “general response to all reviewers” part (Q2).
>
> For the Zero-shot Video QA task, we had a hard time finding baselines that are not trained on the Video QA task. It is unfair to directly compare our method with methods that are trained on a specific task. We compared with JustASK because it is the state-of-the-art Video QA method. However, JustASK is like most other methods that is trained on the Video QA dataset.
>
> For the grade school math task, we would to clarify GPT-FT is directly trained on the GSM8K dataset for generating math solutions. In our method, we used a pre-trained question-solution binary classifier (provide a binary score to evaluate an input question-solution pair match or not). Both GPT-FT and the question-solution binary classifier are trained on the GSM8K dataset, thus it is a fair comparison. Our method of composing pre-trained models still outperforms GPT-FT.
>
> For the image generation task, we added the Efficient-VDVAE [1] baseline. This is a recently proposed unconditional hierarchical VAE model trained specifically on the ImageNet dataset. As illustrated in Table A3, despite Efficient-VDVAE is explicitly trained on ImageNet, it is much worse than our approach (Efficient-VDVAE: KID 23.007 vs. Ours: KID 3.766). More results have been added to Appendix C.4 in the updated paper.
>
> For the robot manipulation task, we add a comparison to the CLIPort model [2] used in Socratic Models [3]. In Socratic models, the evaluation of robot manipulation consists of two steps: 1) a GPT model translates a text goal into a set of subgoals and 2) a CLIPort model executes each subgoal. In contrast, in our robot manipulation task, 1) a vision-language model is used to translate an image goal into a set of subgoals and 2) an iterative MPC procedure is used to execute each given subgoal. Thus a fair comparison is to compare the iterative MPC procedure used in our approach and the CLIPort model used in Socratic models. We thus compare our method to the multilingual CLIPort model on the performance of executing a single relation subgoal. CLIPort model obtains a success rate of 22.6% in this setting, while our approach obtains a success rate of 76.3%.
>
> These results have been added to Appendix C.4 in the updated paper.
>
> [1] Efficient-VDVAE: Less is more
>
> [2] CLIPort: What and Where Pathways for Robotic Manipulation
>
> [3] Socratic Models: Composing Zero-Shot Multimodal Reasoning with Language
>
> **Q3) Possibly Wrong/Confusing Illustrations. a). In Figure 2c, how does the input video frame feed into GPT-2? b) The explanation of LCLIP in Video question answering (VQA) of Sec3.1 is so confusing. I cannot really understand until I go back to read ZeroCap.**
>
> Thanks a lot for pointing out the typo in Figure 2C. We have updated Figure 2 in the paper.
> Thanks for letting us know that LCLIP is not very clear. We have added more explanation of LCLIP in Appendix A.2.

---

> ### Author Response · Authors · 2022-12-08
> **Message to Reviewer kbow**
>
> Dear Reviewer,
>
> Thank you very much for your valuable comments and feedback. We appreciate that you think the idea of using scorers to guide generators to reach a consensus is interesting and can be generalizable to different tasks.
>
> We have addressed your questions in our response. We also have added a baseline for both the image generation and robot manipulation tasks. Please let us know if you have any additional questions. We are happy to clarify or provide additional experiments.
>
> Thanks,
> Authors

---

### Author Response · Authors · 2022-11-18
**General Response (1/2)**

We thank the reviewers for their helpful comments and feedback. All the reviewers find our framework useful for zero-shot multimodal tasks and all reviewers think the idea of using scorers to guide generators to reach a consensus is interesting. Reviewer To1A and Reviewer BdG8 further think the zero-shot results are quite impressive.

We have provided the answer for some shared questions, such as differences from prior work and lack of comparisons to prior work, below. We addressed the rest of the questions in the response to each individual reviewer and have updated the draft based on the reviewers’ comments (changes are highlighted).

**Q1: Difference from previous methods, such as VQGAN-CLIP and ZeroCap (Reviewer kbow and BdG8).**

We would like to clarify that in our paper, we seek to propose a general approach that can *combine different large pretrained models* together to *solve different multimodal tasks* through the principles of *iterative consensus*. While the underlying optimization procedure of instances of our approach may be similar to prior work (VQGAN-CLIP and ZeroCap), our aim is to illustrate that such an optimization principle can be generally applied across a broad class of models and tasks (as shown in the new tasks of mathematically reasoning and robotic manipulation) and can significantly improve the performance by composing ensembles of different models.

One of the key ideas of our paper is *“composing pre-trained models”* to solve multimodal various tasks. In the image generation setting, we illustrate how the combination of CLIP, Classifier Free, and classifier guidance can enable better conditional image generation. In contrast, VQGAN-CLIP focuses on using a single CLIP model to guide the generation of artistic images. Similarly, in the video captioning setting, we illustrate how the combination of different CLIP models enables better video question answering results, while ZeroCap focuses on using a single CLIP model to generate image captions.

We further note that the high-level goal of our paper and VQGAN-CLIP and ZeroCap are completely different. The aim of our paper is to leverage and combine the strengths of large pre-trained models. Building a unified framework for composing multiple models from different domains and taking full usage of these models is extremely important. We are the first to successfully compose multiple pre-trained models with closed-loop feedback and that can solve a variety of multimodal problems in a zero-shot manner (see Introduction Section Paragraphs 2-4). In addition, the scorers can be learned at different times on different data in an incremental-learning manner, enabling the combination of incrementally learned knowledge. Previous methods such as VQGAN-CLIP and ZeroCap cannot solve these challenges and instead focus on narrow tasks of either generating images given language or captioning static images.

We have added this discussion in the paper in Appendix A.5.

**Q2: Lack of comparison with previous works (Reviewer kbow, dTDv)**

We did not compare with many external baselines as most approaches are directly trained to solve the task instead of applied in a zero-shot manner. For example, when we compose GPT-2 and CLIP to do video question answering, neither models is trained on video data, but their combination can be used for solving video tasks.  In this rebuttal, we have added one additional baseline in the image generation and robot manipulation tasks.

For the image generation task, we added the Efficient-VDVAE [1] baseline. This is a recently proposed unconditional hierarchical VAE model trained specifically on the ImageNet dataset. As illustrated in Table A3, despite Efficient-VDVAE is explicitly trained on ImageNet, it is much worse than our approach (Efficient-VDVAE: KID 23.007 vs. Ours: KID 3.766). More results have been added to Appendix C.4.

For the robot manipulation task, we add a comparison to the CLIPort model [2] used in Socratic Models [3]. In Socratic models, the evaluation of robot manipulation consists of two steps: 1) a GPT model translates a text goal into a set of subgoals and 2) a CLIPort model executes each subgoal. In contrast, in our robot manipulation task, 1) a vision-language model is used to translate an image goal into a set of subgoals and 2) an iterative MPC procedure is used to execute each given subgoal. Thus a fair comparison is to compare the iterative MPC procedure used in our approach and the CLIPort model used in Socratic models. We thus compare our method to the multilingual CLIPort model on the performance of executing a single relation subgoal. CLIPort model obtains a success rate of 22.6% in this setting, while our approach obtains a success rate of 76.3%.

These results have been added to Appendix C.4 in the paper.

[1] Efficient-VDVAE: Less is more

[2] CLIPort: What and Where Pathways for Robotic Manipulation

[3]  Socratic Models: Composing Zero-Shot Multimodal Reasoning with Language

---

> ### Author Response · Authors · 2022-11-18
> **General Response (2/2)**
>
> **Q3: Selecting the best scorer or merge scores using their importance weights (Reviewer dTDv and BdG8)**
>
> The key idea of our method is to compose ensembles of pre-trained models, and the way to combine them can be varied, such as adding all the scores as we used in the paper, or taking the best scorer as suggested by the reviewer.
>
> We add two additional experiments in this rebuttal: (1) using the best scorer (scorer that provides the highest score) and (2) using the weighted scores and add them together. Our method composes pre-trained models without training or finetuning, thus we did not learn separated weights for different models. But instead we add an experiment in this rebuttal that uses the scorers (after softmax) of each scorer model as their weights and then composes the scorers using the weighted summed score.
>
> |  |  | Inception Score ↑ | FID ↓ | KID ↓ |
> | ----- | ----- | ----- | ---- | ----|
> | PIC (G+E1+E2) Sum | GLIDE & CLIP + CLS | **30.438** | **29.543** | **5.435** |
> | PIC (G+E1+E2) Max | GLIDE & CLIP + CLS | 24.782 | 30.657 | 6.040 |
> | PIC (G+E1+E2) Weighted | GLIDE & CLIP + CLS | 24.728 | 30.669 | 6.420 |
>
>
> We compare these two baselines with our method that uses the summation of all scores. As shown in the results, using a summed score still generates the best results. The results have been added in Appendix C.3.
>
> **Q4) In industry use cases the GPU memory would overflow to fit all the models simultaneously (Reviewer dTDv and BdG8)**
>
>  In all of our experiments, the ensemble of models can be fitted into a single GPU. For example, we compose the generator model and three scorer models for image generation, it takes 4.7G GPU memory.
>
> In addition, our framework does not require running all the models at the same time. The models can be sequentially sent to GPU and the generated results of each model can be merged after running all the models. Thus our framework can still be applied to industry usage that may require composing lots of pre-trained models.

---

### Decision · Program_Chairs · 2023-01-20

**Decision:**

Accept: poster

**Justification For Why Not Higher Score:**

The paper has some issues regarding metrics and comparisons (explained above). So, it requires some work to become a solid paper.

**Justification For Why Not Lower Score:**

The community benefits from having this paper in the conference, so I voted for acceptance.

**Metareview: Summary, Strengths And Weaknesses:**

The idea of the paper is to use different pre-trained models to complement the capabilities of each other. For example, GPT-3 is a powerful language model but it does not have the capability of visual understanding. On the other hand, methods such as DALL-E generate realistic images but they fail to capture complex language prompts. The paper proposes to ensemble these different pre-trained models to complement each other. A set of proposals are generated by these models, which are later “scored” by other models. There is a closed-loop optimization strategy that encourages the generators to generate better proposals in the next iterations.

Strength:
- The idea of iterative feedback to generate better proposals is interesting.
- It is a big advantage that no fine-tuning is required.
- Zero-shot results are quite encouraging.

Weaknesses:
- Lack of automatic evaluation metrics.
- Lack of comparisons with SOTA methods.


**Note From Pc:**

if the above contains the word "oral" or "spotlight" please see: "oral" presentation means -> notable-top-5% and "spotlight" means -> notable-top-25%. As stated in our emails, we are disassociating presentation type from AC recommendations

**Summary Of Ac-Reviewer Meeting:**

The paper initially received one accept and three borderline ratings. The main concerns were: (1) Lack of novelty with respect to previous works such as VQGAN-CLIP and Zero-Cap, (2) Lack of comparison with other similar works such as Socratic Models or state-of-the-art models for each task, (3) Lack of a reproducible evaluation methodology.

The AC and the reviewers discussed the paper extensively in a video call. Regarding (1), the reviewers agreed that despite the similarity, this paper still has merit to be published. Regarding (2), the reviewers agreed that the Socratic Models work is new and unpublished, and we should not penalize the authors for lack of comparison. Regarding (3), it is a valid concern, and we encourage the authors to develop metrics that can be computed automatically instead of AMT. Otherwise, future work will not be able to provide a comparison.

All in all, the reviewers agreed that the community will learn something new from reading this paper although it has some issues. The AC is in agreement with the majority of the reviewers and recommends acceptance.